# Vegetation history and palaeoclimate at Lake Dojran (FYROM/Greece) during the Late Glacial and Holocene

Alessia Masi[1], Alexander Francke[2,3], Caterina Pepe[1], Matthias Thienemann[2], Bernd Wagner[2], and Laura Sadori[1]

[1]Department of Environmental Biology, Sapienza University, Rome, Italy
[2]Institute of Geology and Mineralogy, University of Cologne, Cologne, Germany
[3]Wollongong Isotope Geochronology Laboratory, School of Earth and Environmental Sciences, University of Wollongong, Wollongong, Australia

*Correspondence to:* Alessia Masi (alessia.masi@uniroma1.it)

**Abstract.** A new high-resolution pollen and NPPs (Non-Pollen Palynomorphs) analysis has been performed on the sediments of Lake Dojran, a transboundary lake located at the border between Greece and Former Yugoslav Republic of Macedonia (FYROM). The sequence covers the last 12500 years and provides information on vegetational dynamics of the Late Glacial and Holocene for southern Balkans. A robust age-model, sedimentological, diatom, and biomarker analyses published previously have been the base for a multi-perspective interpretation of the new palynological data. Pollen analysis revealed that the Late Glacial is characterized by steppic taxa with prevailing Amaranthaceae, *Artemisia* and Poaceae. The arboreal vegetation starts to rise after 11500 yr BP, taking a couple of millennia to be definitively attested. Holocene vegetation is characterized by the dominance of mesophilous plants. *Quercus robur* type and *Pinus* are the most abundant taxa followed by *Quercus cerris* type, *Quercus ilex* type and *Ostrya/Carpinus orientalis*. The first attestation of human presence can be presumed at 5000 yr BP for the contemporary presence of cereals, *Juglans* and *Rumex*. A drop of both pollen concentration and influx together with a $\delta^{18}O_{carb}$ shift indicates increasing aridity and precedes clear and continuous human signs since 4000 yr BP. Also a correlation between *Pediastrum boryanum* and fecal stanol suggests that the increase of nutrients in the water is related to human presence and pasture. An undoubted expansion of human-related plants occurs since 2600 yr BP when cereals, arboreal cultivated and other synanthropic non-cultivated taxa are found. A strong reduction in arboreal vegetation occurred at 2000 yr BP, when the strong Roman Empire impacted on a landscape undergoing climate dryness in the whole mediterranean area. In recent centuries the human impact still remains high but spots of natural vegetation are preserved. The Lake Dojran multi-proxy analysis including pollen data provide a clear evidence of the importance of this approach in the palaeoenvironmental reconstruction. Cross interpretation of several proxies allows to comprehend past vegetation dynamics and human impact in southern Balkans.

# 1 Introduction

High-resolution terrestrial archives from lakes and caves are the basis for understanding past climate changes and vegetation dynamics. Southern Europe counts a large number of extant and paleo-lakes in comparison to northern African and near Eastern regions. This is due to geological features, geographical location and related hydrological and climatic conditions that characterize the southernmost European countries.

While a lot of Spanish, French, Italian and Greek lacustrine records have been studied since the '70s, vegetation dynamics of central Balkans remained for a long time under-investigated due to a variety of natural and historical/political circumstances. Nevertheless, there are two remarkable reasons that make the area important for palaeoenvironmental investigation: 1) the possibility to obtain long lacustrine records due to the high abundance of tectonic lakes, such as Dojran, Prespa, Ohrid, Maliq and 2) the long lasting human occupation. Only recently, the scientific community has appreciated all the potentialities of the area, quite rich in natural lakes, and a number of international teams is currently carrying out interdisciplinary investigations in the area. This massive boost involved either the totally new or the renewed study of several long Quaternary lacustrine sediment records: Ioannina basin (Tzedakis, 1994; Lawson et al., 2004; Roucoux et al., 2011), Lake Ohrid (Wagner et al., 2009, 2017; Lézine et al., 2010; Sadori et al., 2016b), Lake Prespa (Panagiotopoulos et al., 2014), and Tenaghi Philippon (e.g., Tzedakis et al., 2006; Milner et al., 2012; Pross et al., 2015).

The area in general, and the central-western Balkan lakes in particular (Butrint, Dojran, Ohrid, Prespa and Shkodra) are characterized by pristine nature and are important hotspots of biodiversity for aquatic and wetland life, but also for terrestrial plants and animals (Griffiths et al., 2002; Keukelaar et al., 2006; Bojovic et al., 2016). Since the publication of the paper by Willis (1994) on Balkan vegetation history, many new data became available. In particular, investigations of Holocene changes in the last twenty years concerned two lagoons from Mljet island (Jahns and van den Bogaard, 1998), Lake Shkodra (Zanchetta et al., 2012; Sadori et al., 2015a; Mazzini et al., 2016), Lake Butrint (Morellon et al., 2016), Lake Orestiás (Kouli, 2007; Kouli and Dermitzakis, 2010), Lake Voulkaria (Jahns, 2005), several Bulgarian mountain lakes (Tonkov and Marinova, 2005; Tonkov et al., 2008, 2016), the peat bog Vodniza (Tonkov et al., 2018) and Lake Dojran (Athanasiadis et al., 2000).

Athanasiadis et al. (2000) provided pollen results for littoral cores from Lake Dojran covering about the last 5000 years. The lacustrine record appeared to be an important archive of past environmental history and climate change of the region. The high potentiality of the archive appeared even clearer in a new longer record from the central part of the lake. This core spans the Late Glacial and the Holocene and provides a multidisciplinary dataset including geochemical (Francke et al., 2013a), diatom (Zhang et al., 2014) and biomarker (Thienemann et al., 2017) data. The new palynological study on Lake Dojran presented here represents the first detailed and continuous Holocene pollen sequence for the region based on a robust chronology and aims at (1) a high-resolution reconstruction of vegetation dynamics of Lake Dojran, (2) evaluation of the ecosystems response to the transition from dry/cold climate of the last glacial to the wet/hot climate of the Holocene interglacial (3), assessment of the human impact (4) comparison with nearby sequences in order to infer vegetation and human-induced similarities/dissimilarities.

## 2 Site, climate and modern vegetation description

Lake Dojran (41° 12' N, 22° 44' E) is a transboundary lake situated at the border between the Former Yugoslav Republic of Macedonia (FYROM) and Greece. The lake is located at 144 m above sea level (a.s.l.) in a karst depression between two mountain chains: Belles/Belasitsa (peaking at 1870 m a.s.l.) on the northeast and Krousia/Krusa (860 m a.s.l.) on the southeast (Sotiria and Petkovski, 2004). Lake Dojran is a relict of the former Pleistocene Lake Peon, which occupied an area of about 127 km$^2$ (Petkovski et al., 2004). The lake has an elliptic shape with a maximum North-South extension of 9 km, and a maximum East-West extension of 7 km. With its 43 km$^2$ of surface, it is one of the smallest lakes in the region. Small rivers, creeks and groundwaters feed the lake. The outlet is an artificial channel, Doiranitis River, which drains into the Axios River. The outlet is active only when the lake water level is high. It depends on rainfall, summer evaporation, and in recent years on the use of water for irrigation (Griffiths et al., 2002; Sotiria and Petkovski, 2004). The result of the overexploitation during the last decades of the past century was a drastic water level drop (water depth reached 4 m in 2001-2002; Sotiria and Petkovski, 2004). To restore the water depth at about 10 m as it was during the '50s, water from nearby aquifers is pumped into the lake (Bojovic et al., 2016).

The regional climate is mainly influenced by the Mediterranean Sea, separated from the lake by the Thessaloniki Plain. The proximity to the sea and the local morphology mitigate the climate that is warmer than in other Macedonian areas (Sotiria and Petkovski, 2004; Popovska et al., 2014). For the period 1961-2000, the mean annual air temperature and precipitation are 14.3 °C and 612 mm, respectively. Precipitation occurs mainly during the mild winters, when mean temperatures of 3.7 °C are recorded. Summers are dry, with mean temperatures of 26.1 °C (Sotiria and Petkovski, 2004).

The modern vegetation of the Balkans is the result of the combined effects of climate, altitude, soil, Mediterranean Sea influence and anthropogenic activity. Mediterranean, Northern and Eastern European floral elements compose the high biodiversity of the area that is evidenced also by a high endemism (Bojovic et al., 2016). The relative proximity to the Aegean Sea (70 km) influences also the vegetation, resulting in the presence of mediterranean taxa. The mediterranean plant communities are characterized by evergreen forests, dominated by *Pinus halepensis* Mill., evergreen *Quercus* (*Quercus coccifera* L., *Quercus ilex* L.) and *Juniperus oxycedrus* L., while deciduous forest include *Quercus pubescens* Willd., *Carpinus orientalis* Mill., *Ostrya carpinifolia* Scop., *Pistacia terebinthus* L., *Fraxinus ornus* L. and *Acer*. Montane forests cover the mountain chains between 700 and 1700 m a.s.l. with *Abies* and *Juniperus*. Above 1700 m, sub-alpine and alpine vegetation prevails (Eastwood et al., 2004).

In the Dojran catchment, plant associations in which montane, mesophilous and mediterranean are either organized in vegetation belts or mixed up are: *Coccifero-Carpinetum orientalis*, *Carpinetum orientalis*, *Quercetum confertae-cerris*, *Orno-Quercetum petraeae*, *Fagetum submontanum*, *Juglando-Platanetum orientalis*, *Carpinetum orientalis-Quercetum coccifera*, *Carpinetum orientalis-Philyrietosum mediae*, *Carpinetum orientalis-Quercetosum confertae* and *Carpinetum orientalis-Quercetum sessiliflorae* (Sotiria and Petkovski, 2004). Sotiria and Petkovski (2004) list also riparian forests dominated by *Salix alba* and *Populus alba* galleries. Athanasiadis et al. (2000) report for the vegetation surrounding Lake Dojran the presence of mesophilous forests mainly composed by *Quercus pedunculiflora* C. Koch., *Fraxinus oxycarpa* Willd., *Alnus glutinosa* (L.)

Gaertn., *Ulmus minor* Mill., *Ulmus laevis* Pall., *Platanus orientalis* L., *Vitis sylvestris* Gmel., *Humulus lupulus* L., *Periploca graeca* L. Evergreen oaks are mainly represented by *Quercus coccifera* L. The lake and part of its surrounding are considered important areas of biodiversity and included in the IPA (Important Plant Areas) list (Melovski et al., 2012). Vegetation nearby shores is typical for pseudomaquis and consists of *Quercus coccifera* L., *Quercus pubescens* Willd., *Carpinus orientalis* Mill.,

*Clematis flammula* L., *Juniperus oxycedrus* L., *Pistacia terebinthus* L., *Ulmus minor* Mill., *Ficus carica* L., *Rhamnus saxatilis* subsp. *rodopea* (Velen.) Aldén (*Rhamnus rhodopea*) (http://plantlifeipa.org/Factsheet.asp?sid=1495, Čarni et al., 2003). Extensive reed beds surround the lake shore. *Phragmites australis* (Cav.) Steud., *Schoenoplectus lacustris* (L.) Palla (=*Scirpus lacustris* L.), *Typha angusifolia* L., *Typha latifolia* L., *Sparganium neglectum* and *Sparganium erectum* aggr. are the most common taxa. When the water level decreases the wet soil becomes a green carpet of the grass *Paspalum distichum* ssp. *paspalodes*

(Michx.) Thell. (Athanasiadis et al., 2000; Sotiria and Petkovski, 2004; Bojovic et al., 2016). Today, anthropogenic activity including cultivation of grapes, vegetables, cereals, tobacco and forage crops seriously disturbed the vegetation surrounding the lake (Bojovic et al., 2016).

## 3  Material and Methods

### 3.1  Coring, chronology and sedimentological analysis

A 717 cm long sediment sequence (Co1260) was recovered by gravity corer and percussion piston corer in June 2011 from the Macedonian part of the lake (Fig. 1), where a hydro-acoustic survey (Innomar SES-2000 compact, 10 kHz) indicated undisturbed, horizontal bedded sediments (Francke et al., 2013a). The age-model is based on 13 calibrated radiocarbon ages derived from terrestrial plant material, charcoal, carbonates and bulk organic C samples (Francke et al., 2013a). The sediment sequence encompasses the last ∼12,500 years, i.e., the Late Glacial and the entire Holocene. Sedimentological, stable isotope

(Francke et al., 2013a), diatom (Zhang et al., 2014) and biomarker (Thienemann et al., 2017) data provide a robust basis for a new study of pollen on the sequence.

### 3.2  Pollen analysis

Palynology has been carried out on 146 sediment samples taken from throughout core Co1260. The mean temporal resolution equals 87 years. Between ca. 12500 and 11400 yr BP, the mean temporal resolution is 71 years, and from ca. 3250 yr BP to 2004

AD it equals 66 years. For each sample, 0.29 to 0.57 g of dry sediment were chemically processed with alternating treatment of HCl (37 %), HF (40 %) and hot NaOH (10 %) (Fægri and Iversen, 1989, modified) in order to remove detrital matter from the sediments. Following Stockmarr (1971), tablets containing a known amount of *Lycopodium* spores were added in order to estimate pollen, NPPs (Non-Pollen Palynomorphs) and microcharcoal concentrations. Pollen concentration has been used to elaborate influx data on the basis of the sedimentation rates as inferred from the existing age model. Pollen and microcharcoal

influx data are an estimation of the amount of pollen grains and charcoal fragments deposited incorporated annually in single unit surface (Berglund et al., 1986). The first is an estimate of the plant biomass, the second of the burnt biomass.

The pollen basis sum is constituted by terrestrial spermatophytes. For other taxa, percentages were calculated on different pollen basis sums as reported by Berglund et al. (1986). The identification of pollen morphology is based on atlases (Reille, 1992, 1995, 1998) and reference pollen collections. Pollen diagrams were drawn against both depth and time scales using TILIA program (Grimm, 1992). All arboreal and non-arboreal pollen taxa with values higher than 2 % of the total pollen sum were used for the CONISS cluster analysis (Grimm, 1992). Pollen zones have been established following CONISS indication and visual inspection.

Pollen grains of several species belonging to genus *Quercus* have been divided in three morphological groups according to Smit (1973): *Quercus robur* type, which comprehends all deciduous oaks; *Quercus cerris* type, which comprehends all semievergreen oaks plus *Quercus suber*; and *Quercus ilex* type, including all the evergreen oaks minus *Quercus suber*. The denomination *Quercus ilex* type was kept even if *Quercus coccifera* is the most common evergreen oak in the area. Among Poaceae, cereals have been identified and mainly comprehend grains ascribable to *Hordeum* group and *Secale cereale* (Andersen et al., 1979). *Triticum/Avena* pollen grains are present as well. Here, we present a cumulative curve for cereals. Cereal type includes also pollen grains of some wild Poaceae. Considering Asteraceae, pollen grains belonging to the two sub-families Asteroideae and Cichorioideae have been identified. As the tribe Cichorieae, European native, is the only one with fenestrate pollen grains (Florenzano et al., 2015) it would be more appropriate name Cichorioideae Cichorieae. We keep anyway the more generic name, as it could be more clear for the reader. Among Algae, species of *Pediastrum*, *P. simplex*, *P. simplex* var. *sturmii* and *P. boryanum* have been recognized following Komárek and Jankovská (2001). Charcoals have been divided in three dimensional groups according to the length of the shorter axis. The particle size reveals the origin of charred fragments: the 10-50 $\mu$m group indicates regional fire, 50-125 $\mu$m group landscape/regional fire and >125 $\mu$m group local fire (Whitlock et al., 2010; Sadori et al., 2015b).

## 4 Results

The pollen concentration spans from ca. 11200 to 692300 grains/g and the state of preservation of pollen is variable. The mean count of terrestrial pollen is 409 grains/sample. A total of 83 terrestrial pollen taxa are identified and comprehend 37 trees and shrubs and 46 herb taxa. Most abundant taxa are: *Quercus robur* type, *Pinus*, *Quercus cerris* type, *Quercus ilex* type and *Ostrya/Carpinus orientalis* among arboreal plants; Amaranthaceae, Poaceae and *Artemisia* among herbs. CONISS and the results of the pollen analyses, plotted as percentage, concentration and influx values, are shown in Figs 2 and 3. In Fig. 4, ecological groups (mesophilous, xeric, pioneer, mediterranean, montane, and synanthropic), geochemical proxies, biomarkers and planktonic diatom curves are plotted and diatom zones description was summarized and simplified (for major details and more precise description see the paper by Zhang et al., 2014). Amaranthaceae are included in the sum of xeric taxa, even if, especially in the last millennia, some of them can be also an indication of human disturbance. Pollen zones are described taking in consideration also sedimentological and stable isotope data (Francke et al., 2013a), diatom data (Zhang et al., 2014), and biomarker analyses (Thienemann et al., 2017).

## 4.1 Pollen zone D-1 (716-599 cm, 12500-11550 yr BP, duration 950 yr)

AP % ranges from 8 to 38 %. Total pollen concentration (pollen grains/g) varies between ca. 11200 and 52600, influx values between ca. 7200 and 14700, and the number of taxa between 14 and 36. The zone is dominated by herbs. Amaranthaceae are overwhelming along the entire zone (28-70 %, see also concentration and influx values in Fig. 3) and peak at 12000 yr
BP (Fig. 2). At the bottom of the record (12500 yr BP), *Artemisia* has its highest values (25-32 %), followed by increasing Poaceae that overtake 30 % at around 11900 yr BP. The presence of 26 arboreal taxa, mainly mesophilous and pioneer ones, is noteworthy. The latter (max. 6 %) are composed by *Ephedra* (0-3 %), *Juniperus* (0-2 %), *Corylus* (0-2 %), Rosaceae undiff. (0-2 %) and *Betula* (0-1 %). *Pinus* pollen shows low values, with two exceptions at 12400 (10 % complete and 14 % of broken grains) and 12300 (6 % complete and 16 % of broken grains) yr BP. *Quercus robur* type (0-14 %) shows a slight expansion
around 12300 yr BP together with a peak of *Alnus* (5 %). A gradual change in herb vegetation, consisting of a decrease in *Artemisia* and an increase in Amaranthaceae and Poaceae, is found after 12300 yr BP. This change matches a low lake level as inferred from hydro-acoustic data and an increase of salinity indicated by diatoms. Algae (mainly *Pediastrum boryanum*), aquatic (*Myriophyllum*) and riparian (*Typha angustifolia*) plants indicate lacustrine conditions. The sediment features indicate cold and dry conditions until 12100 yr BP. Charcoals presence attests that local and regional fires are present since 12000 yr
BP. NPP fungal remains (*Glomus* and ascospores) increase since 11900 yr BP. Although the presence of clay clasts that may have formed under subaerial conditions and thus indicate redeposition of sediments in the lowermost core section (Francke et al., 2013a), the pollen assemblage shows herb vegetation succession typical of the Late Glacial period.

## 4.2 Pollen zone D-2 (593-541 cm, 11500-10950 yr BP, duration 550 yr)

AP % ranges from 31 to 45 %. Total pollen concentration (pollen grains/g) varies between ca. 25100 and 53200, influx values
between ca. 2600 and 5300 and the number of taxa between 34 and 38. This phase represents the onset of Holocene reforestation. The passage is abrupt and marked by trees increasing from 17 % of D-1 to 31 % of the first sample of D-2. Herbs still prevail and arboreal vegetation is always under 45 %. Pioneer taxa (Fig. 4) play a role more important than in the previous zone, with *Ephedra* (>0-4 %) slightly decreasing while *Juniperus* (1-6 %), Rosaceae undiff. (1-6 %) and *Betula* (1-3 %) increase. *Quercus robur* type (5-16 %) and *Quercus cerris* type (2-9 %) dominate among mesophilous taxa with low contribution of other
taxa like *Ostrya/Carpinus orientalis* (1-3 %) and *Ulmus* (1-2 %). Mediterranean vegetation is mainly represented by *Quercus ilex* type (1-2 %). Poaceae (17-25 %) increase and Amaranthaceae (10-30 %) highly decrease. *Artemisia* (3-12 %), Asteroideae undiff. (2-8 %) are still abundant. *Galium* (0-6 %) and Lamiaceae (0-4 %) show the highest values in this zone. Pollen assemblage and $\delta^{18}O_{carb}$ values indicate increasing precipitation. Low winter temperatures are inferred by sporadic occurrences of sand lenses. Low TOC and the abundance of planktonic oligotrophic-mesotrophic diatom indicate low productivity. Charcoal
indicates sporadic local and low regional fires. Poaceae, *Typha angustifolia* type (2-4 %) and Cyperaceae (0-4 %) could have formed riparian vegetation belts along Lake Dojran shores. This vegetation could have trapped clastic matter and nutrients explaining the limited productivity of the lake as registered by the sedimentological investigation and by the oligotrophic to mesotrophic water state as inferred from diatom analyses.

### 4.3 Pollen zone D-3 (533-487 cm, 10850-10100 yr BP, duration 750 yr)

AP % ranges from 36 to 67 %. Total pollen concentration (pollen grains/g) varies between ca. 39500 and 84300, influx values between ca. 3100 and 5900 and the number of taxa between 35 and 52. Several strong and rapid changes of vegetation are registered in this zone. Arboreal vegetation rises rapidly and woodland is dominated by *Quercus robur* type (10-32 %). None of the other mesophilous plants shows significant amounts. The only other taxon that shows a considerable presence is *Pinus* (9-15 %). They are mainly due to Rosaceae undiff. (3-8 %). Among herbs, *Artemisia* (2-5 %) and Asteroideae undiff. (3-6 %) are still present together with Amaranthaceae (4-18 %) and *Galium* (2-5 %). Poaceae (14-22 %) decline even if still showing high percentage and influx values. The increase in water availability is evidenced not only by the rapid increase of mesophilous taxa and the strong reduction of xeric taxa (from 23 to 7 %), but also by the strong expansion of *Pediastrum simplex* (>0-10 %), *Pediastrum simplex* var. *sturmii* (3-63 %), *Pediastrum boryanum* (3-38 %; Komárek and Jankovská, 2001) and by high values of diatom planktonic taxa. At around 10400 yr BP an important change occurred in all algae. *Pediastrum boryanum* increases (up to 38 %) and facultative planktonic diatoms strongly increase both in percentage (up to ca. 60 %) and in concentration. Both data could therefore either indicate the presence of more nutrients or shallowing water. As vegetation suggests increasing wetness, the hypothesis that changes in the lacustrine water could be due to increasing temperature favouring increase of nutrients, seems strongly supported. Mesophilous vegetation shows a continuous increasing trend that can be related to the rise in humidity suggested by $\delta^{18}O_{carb}$ at the end of the zone. Low concentration values of microcharcoals indicate the absence of local fire and sporadic regional ones. Low fire activity is also indicated by biomarkers with low concentration of polycyclic aromatic hydrocarbon (PAHs).

### 4.4 Pollen zone D-4 (483-443 cm, 10150-9400 yr BP, duration 750 yr)

AP % ranges from 62 to 76 %. Total pollen concentration (pollen grains/g) varies between ca. 57900 and 250700, influx values between ca. 3200 and 11600 and the number of taxa between 31 and 49. Mesophilous vegetation dominates in the pollen zone. Deciduous oaks (26-41 %) record a minimum at 9700 yr BP in correspondence with an increase of Poaceae (5-16 %) and semievergreen oaks (2-16 %). Among aquatic plants, *Alisma* (>0-3 %) shows its highest value. Deciduous oaks reduction matches a minimum in planktonic diatoms and in total diatom concentration. Pioneer taxa are on the whole reduced, with Rosaceae undiff. (1-5 %) still important and *Corylus* (0-4 %) increasing. Xeric taxa (*Artemisia*, 1-3 %; Amaranthaceae, 1-5 %; *Juniperus*, 0-3 %) and Asteroideae undiff. (1-4 %) are reduced. Pine is between 7 and 14 %. High values of *Galium* (0-5 %) and significant values of Lamiaceae (0-3 %) are still found, a spread of *Sanguisorba* cf. *minor* (1-7 %) is found too. *Abies* (0-1 %) shows sporadic presence since 9800 yr BP. Microcharcoals are indicating the presence of both regional and local fires. According to the pollen assemblage and sedimentological data, this time period was characterized by increasing humidity in particular during summer. However, diatoms suggest low lake levels.

### 4.5 Pollen zone D-5 (439-385 cm, 9300-7600 yr BP, duration 1700 yr)

AP% ranges from 76 to 93 %. Total pollen concentration (pollen grains/g) varies between ca. 125800 and 342200, influx values between ca. 3700 and 13500 and the number of taxa between 29 and 40. *Quercus robur* type is dominating, with oscillation from 34 to 51 % and high influx values in particular at 9000 yr BP. The other woodland elements (*Quercus cerris* type, 3-15 %; *Ostrya/Carpinus orientalis*, 1-9 %; *Corylus*, 1-6 %), together with the first sporadic traces of *Carpinus betulus* (0-4 %), indicate rising temperature. *Fagus* is more continuously present all over the zone even if with values below 1 %. In general, the rising value of AP % is related to the increase of *Abies* (0-4 %), *Ostrya/Carpinus orientalis* and in particular of *Pinus* that triples its amount along this zone, from 6 to 20 %. On the contrary, none of the mediterranean taxa increases. The rising of pine together with fir suggests that they both come from mountain species organized in altitudinal belts. Denser forest canopy (AP 89 % at 8400 yr BP) matches a wetness increase indicated by lower $\delta^{18}O_{carb}$ and a change in lake level suggested by diatom (from shallow to high stands). *Alisma* (>0-2 %) is attested at the very beginning of this zone. Turbid water inferred from diatoms was probably responsible for decreased *Pediastrum*, as whole changing from 41 % at 8700 yr BP to 3 % at 8200 yr BP. Among herbs, Poaceae (2-9 %) still prevail together with *Sanguisorba* cf. *minor* (1-7 %), but both of them show a decreasing trend.

### 4.6 Pollen zone D-6 (383-315 cm, 7500-4000 yr BP, duration 3500 yr)

AP % ranges from 84 to 94 %. Total pollen concentration (pollen grains/g) varies between ca. 200900 and 692300, the highest value of the record, influx values between ca. 3500 and 12600 and the number of taxa between 25 and 40. Mesophilous vegetation is well attested during mid-Holocene. The arboreal pollen percentages are always around 90 %. *Quercus robur* type dominates (35-61 %) followed by *Pinus* (16-31 %), *Quercus cerris* type (4-15 %), *Abies* (2-9 %) and *Ostrya/Carpinus orientalis* (1-6 %). Cereals show some intermittent presence since 4900 yr BP. Montane taxa are quite well represented, in particular *Abies*. Mediterranean vegetation is showing minimum values. Palynological data indicate a stable climate with a high lake level and humid conditions, which is confirmed by diatom, sedimentological and $\delta^{18}O_{carb}$ data. Relatively stable conditions are also visible from the biomarker records. In addition, diatoms point out a eutrophic state of the lake. The littoral vegetation belt was limited as indicated by the low amount of Poaceae (2-9 %). Other herbs show low values. Cereals (>0 %) show a continuous presence between 5000 and 4800 yr BP. In this zone, *Pediastrum* has still low values related to the permanence of turbid water. Presence of different sizes of charcoal attests the presence of regional and local fires.

### 4.7 Pollen zone D-7 (311-257 cm, 3900-2650 yr BP, duration 1250 yr)

AP % reach the highest values in this zone, it ranges from 82 to 95 %. Total pollen concentration (pollen grains/g) varies between ca. 90200 and 315200, influx values between ca. 5100 and 13700 and the number of taxa between 26 and 42. Pollen data register a rapid increase of *Pinus* (21-35 %), reaching the highest values of the whole sequence, and decreased values of *Quercus robur* type (29-46 %), anyway always dominating. *Fagus* presence (>0-3 %), among montane taxa slightly increasing, is now definitively attested after a very slow increase since ca. 6000 yr BP. Other arboreal plants are *Quercus cerris* type (4-22 %), *Abies* (0-10 %) and *Quercus ilex* type (>0-6 %). Gradual environmental changes can be addressed for the increase

of mediterranean (mainly due to evergreen oaks) and xeric taxa. Since 3000 yr BP, the latter show values higher than 5 %. The occurrence of such high values was recorded at 9900 yr BP for the last time. Cereals (<1 %) are almost continuously present since 3600 yr BP. Planktonic diatoms strongly decrease soon after 3000 yr BP, indicating lake-shallowing. Curves of *Pediastrum* taxa fluctuate, showing increases since the bottom of the zone D-7, suggesting water eutrophication. In particular, *P. boryanum* matches the fecal stanol (produced in the gut of mammalians) record, suggesting a correspondence between humans, pasture and increase of nutrients in water. Geochemical data attest distinct changes in some proxies, suggesting a great instability. In particular the C/N ratio shows a significant increase of nutrients at 3200 yr BP. At the same time, clastic matter deposition starts to increase suggesting intensive erosion in the catchment.

### 4.8 Pollen zone D-8 (253-217 cm, 2600-2000 yr BP, duration 600 yr)

AP% ranges from 83 to 90 %. Total pollen concentration (pollen grains/g) varies between ca. 75200 and 383300, influx values between ca. 4400 and 25800 and the number of taxa between 28 and 44. An important vegetation change is suggested by the rapid increase of pioneer taxa mostly due to *Juniperus* (>0-8 %). This is indicating deterioration of the forest ecosystems even if AP % is only slightly reduced. *Abies* (2-7 %) begins to decline. *Quercus robur* type (25-39 %), and *Pinus* (22-35 %) are dominating and accompanied by *Quercus cerris* type (4-14 %), *Quercus ilex* type (0-7 %), *Fagus* (>0-4 %), *Ostrya/Carpinus orientalis* (1-3 %). Among arboreal plants *Olea* (0-2 %), *Juglans* (0-1 %) and *Castanea* (0-1 %) appear to be more continuous than in previous zones. These three taxa are sporadically present since the early Holocene but show a clear increase since around 2500 yr BP. *Olea* is the first taxon to increase, followed soon after by *Juglans* and then by *Castanea*. Their contemporary and increased presence can be taken as an evidence of cultivation (Mercuri et al., 2013a). This hypothesis is supported by the expansion of cereals (reaching 4 %) and the sporadic presence of *Plantago* since the bottom zone. The pollen assemblage suggests the use of agricultural practises. This hypothesis is confirmed by increased *Glomus*, suggesting soil erosion. Charcoals attesting regional fires are documented all over the zone. Diatoms assemblage evidences shallow water and eutrophic conditions, which is confirmed by similar trends in fecal stanols and *Pediastrum boryanum*. At 2400 yr BP, the expansion of *Pediastrum simplex* var. *sturmii* could be an indication of increasing water temperature.

### 4.9 Pollen zone D-9 (213-161 cm, 2000-1400 yr BP, duration 600 yr)

AP% ranges from 65 to 81 %. Total pollen concentration (pollen grains/g) varies between ca. 33400 and 126200, influx values between ca. 2800 and 9600 and the number of taxa between 40 and 50. At the beginning of the zone, the arboreal vegetation rapidly drops due to the abrupt decline of *Pinus* (from 18 to 5 %, from 50000 to 17000 pollen grains/g) accompanied by a slower reduction of *Abies* (from 7 to 1 %, from 10000 to 1000 pollen grains/g). Even not at similar amplitude, there are quite clear similar patterns in many taxa both in concentration and influx data. Most taxa, including oaks, appear to be affected by a strong biomass decrease, even if it occurs slightly earlier than in the two mentioned conifers. The decrease started at the end of zone D-8 for *Quercus robur* type (38 to 26 %, from 113000 to 37000 pollen grains/g) and *Quercus cerris* type (from 7 to 6 %, 113000 to 37000 pollen grains) that reach lower values ca. 50 years before the drop of pine and fir. Pioneer, xeric and synanthropic taxa increase in the zone. Human presence is indicated by the high amount of synanthropic plants such as

cultivated (e.g. cereals, 0-3 %; *Castanea*, 0-1 %; *Olea*; *Juglans*) and ruderal ones (*Plantago*, 0-2; *Rumex*, 0-2; *Urtica*, >0-2). Forest clearance and the relative high amount of xeric taxa, mainly *Artemisia* (1-2 %), can be either the consequence of increased aridity or human impact. The increase of Poaceae (7-15 %) and Amaranthaceae (2-5 %) could be due to low lake stands and the relative enlargement of shallow areas covered by reed beds. The productivity in the lake seems to be rather important as evidenced by TOC/TN and diatom assemblages. Charcoal assemblage shows that regional and local fires are of scarce importance.

## 4.10 Pollen zone D-10 (153-0 cm, 1250 yr BP-present, duration 1200 yr)

AP% ranges from 60 to 86 %. Total pollen concentration (pollen grains/g) varies between ca. 55000 and 230900, influx values between ca. 6500 and 25600 and the number of taxa between 32 and 55. Mesophilous taxa drop. This change is mainly due to decreasing *Quercus robur* type (8-39 %) and *Quercus cerris* type (3-16 %), while *Carpinus betulus* show moderate alternations with values (>0-4 %) as high as in OD-5. Mediterranean taxa increase, peaking in the last samples. This increase is mainly due to *Quercus ilex* type (1-19 %), but also to *Olea* (0-4 %). *Pinus* (7-26 %) well recovered and reaches 26 % at about 500 yr BP. *Abies* remains below 2 %, suggesting that higher-altitude environments do not easily recover, being more fragile. Xeric (mainly Amaranthaceae, 0-11 %) and synanthropic taxa increase along the whole zone, in particular since 500 yr BP. *Artemisia* show rather high values in the second part of the zone. Amaranthaceae, included in the sum of xeric taxa, can comprehend also ruderal herbs. The proximity of the lake is colonized by arboreal riparian vegetation, mainly *Alnus* (max 2 %). Poaceae could form a *Phragmites* belt around the lake like today. Shallow lake conditions are inferred from diatoms and *Myriophyllum*. This water plant generally grows in water up to 2 m deep (Azzella and Scarfó, 2010). Increasing temperature and eutrophic water have been also evidenced by the rising of *Pediastrum* taxa. Bottema et al. (1974) associated the increase of *Pediastrum simplex* to the soil fertilization due to cattle. In the same period biomarkers (especially the fecal stanol record) indicate strong anthropogenic impact in the catchment. Concentration and influx values of charcoal are the highest of the diagram for regional fires between 1000 and 1300 yr BP. The last part of the pollen record reflects the increasing human impact and the consequences of the recent tendencies in the shaping of landscape around the lake.

## 5 General discussion and multi-proxy comparison

The Lake Dojran pollen sequence starts during the Late Glacial with the dominance of xeric taxa. Most arboreal plants are present at the bottom of the core, suggesting the presence of glacial refugia for montane and mesophilous taxa in the catchment of Lake Dojran. In particular *Pinus*, *Quercus robur* type and *Ostrya/Carpinus orientalis* peaks are recorded before 12000 yr BP. Fluctuating presence of coniferous and deciduous taxa in glacial periods is recorded in several lakes of the central-eastern Mediterranean: Prespa (Panagiotopoulos et al., 2013), Ohrid (Sadori et al., 2016b), Ljubljana (Willis, 1994), Ioannina (Bottema et al., 1974; Tzedakis, 1994), Bulgarian mountain lakes (Tonkov and Marinova, 2005; Tonkov et al., 2008, 2016), Monticchio (Allen et al., 2009), Vico (Magri and Sadori, 1999), Pergusa (Sadori and Narcisi, 2001), Trifoglietti (De Beaulieu et al., 2017), Van (Litt et al., 2009).

The onset of Holocene reforestation is dated at Dojran to 11500 BP and infers increasing humidity and temperature. The data shows the decline of steppe-related plants and also the presence of pioneer trees as Rosaceae, *Juniperus* and *Betula* that constitute forest patches accompanied by *Pinus* and by all oak types. This behaviour is typical of the succession from glacial to interglacial phases in southern Europe (Tzedakis, 2007) as shown in Fig. 5 where some of the available records are shown.

Similarly to other lacustrine sites (Prespa, Panagiotopoulos et al., 2013; Ioannina, Lawson et al., 2004 and Pergusa, Sadori and Narcisi, 2001) a couple of millennia are needed at Dojran to achieve 80 % of AP in zone D-5 at ca. 9300 yr BP showing fairly resilient ecosystems. In particular, rising humidity is evidenced, as for Lake Van (Litt et al., 2009), by the increasing trend of deciduous (and semideciduous) oaks. In pollen zone D-2 (ca. 11500-10900 yr BP) AP trend shows anyway a first start soon followed by a decrease of percentage in mesophilous taxa that deserves consideration. It is due to a decrease of *Quercus*

*robur* type partly balanced by an increase of *Q. cerris* type (peaking at 11100 yr BP). This shift between the two oak types was probably a signal of a temporary reduction of humidity, a sort of short-term stasis of forest expansion. A similar pattern (evidenced in Fig. 5) is observed at Prespa (Panagiotopoulos et al., 2013) and Ioannina (Lawson et al., 2004) in the Balkans, and at Vico (Magri and Sadori, 1999) and Pergusa (Sadori and Narcisi, 2001) in Italy. It is likely to be synchronous and secular differences can be attributed to uncertainties of each site's age model. The pattern of zone D-2 seems to be repeated once more

at Dojran in zone D-4 (10200-9400 yr BP), where a similarity in mesophilous taxa and TOC curve trends is found (Fig. 4). This forest opening is probably less pronounced at other sites, but it matches an important change detected at higher elevations from "wetter" to "dryer" taxa. At lake Trilistnika (Tonkov et al., 2008) in fact, after a short and sharp increase of *Abies*, soon followed by *Quercus robur* type, *Quercus cerris* type prevails.

The forest is definitely established at ca. 9300 yr BP (beginning of zone D-5) with AP at 80 %, increased pollen concentration

and influx values and relatively increasing TOC (Fig. 3 and 4; Francke et al., 2013a). The rising AP% is paralleled by a decreasing trend in the average chain length of vascular plant n-alkanes, also indicating increasing arboreal vegetation (Thienemann et al., 2017). Pioneer taxa, with the exception of wet-demanding *Corylus* that is increasing, and the light-demanding *Sanguisorba* cf. *minor* were replaced first by *Abies* and then at ca. 8500 yr BP by *Ostrya/Carpinus orientalis* and *Carpinus betulus*. This forest succession, resulting in increase of "other mesophilous" and montane taxa (Fig. 4) matches enhanced humidity ($\delta^{18}O_{carb}$)

and high lake level inferred from diatoms. The latter is also confirmed by decreasing Poaceae, which may indicate a reduction of the lacustrine vegetation belt formed by *Phragmites* and/or *Paspalum*. The increase of humidity probably allowed the colonization of higher altitudes and resulted in the definitive attestation of vegetation belts. Even if temperate deciduous forest prevails, mountain taxa such as *Abies* and *Fagus* probably partly replaced pioneer ones on the two mountain chains (Belles and Krousia) surrounding the lake. Arboreal vegetation (and deciduous oaks above all) prevails not only at Lake Dojran but also at

other regions in the eastern and central Mediterranean, such as at Lake Prespa (Panagiotopoulos et al., 2013), in the Ioannina basin (Lawson et al., 2004), at Tenaghi Philippon (Müller et al., 2011), at Lake Iznik (Miebach et al., 2016), at Eski Acigöl (Woldring and Bottema, 2003), at Lake Van (Litt et al., 2009) and at Lake Pergusa (Sadori and Narcisi, 2001; Sadori et al., 2013).

The pollen assemblages of Lake Dojran show no clear evidence of the 8.2 ka cooling event. The zone shows rather stable

vegetation conditions and very low sedimentation rate. The event is considered the most prominent and abrupt climate change

at northern latitudes of the entire Holocene (Johnsen et al., 2001). Only some of the mediterranean pollen records (Lake Maliq: Bordon et al., 2009; Tenaghi Philippon: Pross et al., 2009) register the 8.2-event, usually consisting in a more or less pronounced short phase with reduced precipitation (e.g., Staubwasser and Weiss, 2006; Kotthoff et al., 2008; Pross et al., 2009; Göktürk et al., 2011; Miebach et al., 2016). It is interesting to note that hydro-acoustic data indicated a low lake level centred at 8.2 ka (Francke et al., 2013a) supported by a reduction of Poaceae. Water could be partially trapped in wider or longer persisting snow cover of the mountains. Amount of mediterranean plants is low indicating that the water availability is related to temperate climate and the abundance of mesophilous plants indicates wet conditions. Pross et al. (2009) suggest a thermal gradient between inland and coastal settings in the Aegean Sea with weaker winter cooling at the coast. The connection of the Dojran area with Mediterranean Sea could have mitigated the temperature reduction that, together with the resilience of vegetation, could have masked the 8.2 event impact.

Traces of synanthropic plants, including cereals, *Juglans* and *Rumex*, are found since 5000 yr BP. Biomarker data also possibly indicate first human activities around 4500 yr BP (Thienemann et al., 2017). Athanasiadis et al. (2000) found clear signs for first human activity at that time in two cores from the eastern edge of the lake. Co1260 (this article) was recovered from a central position in the lake and records the onset of human settlements less pronounced, probably due to a higher distance from these settlements. However, the presence of cereals can be taken as an indication of cultivation in the area that was populated since the early Neolithic, due to the migration of populations from Anatolia and nearby Greece performing livestock farming (Kaiser and Voytek, 1983). A wide range of cereals and legumes are attested in the archaeological excavation of Anza (ca. 6500-5000 yr BC) in FYROM for the Neolithic (Gimbutas, 1974). In the nearby Struma Valley (Bulgaria) the size and number of archaeological sites increase, indicating a strong increase of population particularly in the second half of the Late Neolithic (5200-4900 yr BC, Marinova et al., 2012). Archaeological remains evidence a new increase of sedentarism during the Bronze Age, with increasing agriculture practises (Kokkinidou and Trantalidou, 1991). In Lake Dojran sediments, *Juglans* is attested for the first time at around 7000 yr BP in agreement with a recent paper of Pollegioni et al. (2017) setting that the first exchanges of germplasm between Near East and Aegean region are dated at 6th millennium BP. Contemporary presence is sporadically recorded at Lake Ribno Banderishko (southwestern Bulgaria, Tonkov et al., 2002) while major quantities are found at Orestiás (Kouli, 2007). On the contrary, at Shkodra, Prespa and in the previous data from Dojran, *Juglans* is attested only since 3500 yr BP in more significant quantities. The early presence of walnut in the Dojran record can be related to the economic exchange that would have spread the plant from Near East through Turkey to Europe. Walnut tree is in fact precious not only for the edible fruit but also for the valuable timber.

At ca. 4000 yr BP, a sharp influx and concentration drop matches and increase of *Pinus* marking the start of zone D-7, suggesting decreasing humidity. AP % remains rather stable, probably due to the expansion of *Pinus*. This change of forest composition probably favoured the slight increase of *Fagus*. The other proxies available at Dojran, which are not affected by human impact, suggest aridification and lower lake levels too. Thus $\delta^{18}O_{carb}$ yield the highest values of the last 9500 yr, and planktonic diatoms decrease strongly. Biomarker data also indicate a drier climate around 4000 yr BP. Forest opening, occurring soon before and culminating at 4000 yr BP, is indicated in most pollen records shown in Fig. 5 pointing out that the phenomenon is present both in Italian and Balkan peninsula and is strictly related to regions under mediterranean influence. In fact aridification is missing

both at Lake Van (Litt et al., 2009) and in the Central Rila mountains in Bulgaria where *Picea* shows an increasing trend since 5000 yr BP (Tonkov et al., 2016). Moreover, it is still under debate if this change is caused by climatic change or human-induced. In central Italy, a decrease in humidity, detected soon before 4000 yr BP, is found in low-stand lake levels (Giraudi et al., 2011) and in speleothems (Zanchetta et al., 2016). This change in hydrology and seasonality (longer summer drought)

surely affected forest plants, especially mesophilous taxa. Animal husbandry, cultivation and metallurgy were probably intensifying and accelerating the phenomenon (Sadori et al., 2004) causing forest clearance all over the central mediterranean area (Denéfle et al., 2000; Jahns, 2005; Caroli et al., 2007; Sadori et al., 2008; Bordon et al., 2009; Di Rita and Magri, 2009; Tinner et al., 2009; Combourieu-Nebout et al., 2013; Mercuri et al., 2012, 2013a). A major role is to be ascribed also to fire, which increased in the whole mediterranean area (Vanniére et al., 2011; Sadori et al., 2015b). According to some authors, the use of

fire and the anthropogenic degradation of forest (Marinova et al., 2012) could have favoured the spread of *Fagus*, but a climate driven cause should also to be taken into account (Giesecke et al., 2007; Valsecchi et al., 2008).

Since 2600 yr BP, Lake Dojran pollen assemblage shows a considerable expansion of anthropogenic indicator taxa (the synanthropic non-cultivated and cultivated plants described by Behre et al., 1990). Littoral cores studied by Athanasiadis et al. (2000) record primary and secondary anthropogenic indicators (mainly *Triticum* type and *Secale*) since ca. 3500 yr BP with

lower percentage. Another major vegetation change around this date is an expansion of *Juniperus* that probably replaced the mesophilous forest disturbed by human activity. Also several arboreal cultivated taxa increase in zone D-8: *Castanea*, *Juglans* and *Olea*. Mercuri et al. (2013a) introduced the OJC (*Olea*, *Juglans*, *Castanea*) sum to highlight the increasing human activity in the mediterranean area, as these trees increase from the Bronze Age onwards all over Italy. Mercuri et al. (2013b) made a step forward adding seven selected anthropogenic non arboreal pollen indicators (APi) common in archaeological sites to

investigate the shaping of the cultural landscape (Marignani et al., 2017).

At Lake Dojran, cereals percentages are low, but quite significant (around 5 %) to attest a well-developed agricultural system (Sadori et al., 2016a). This is due to the self-pollination and the consequent under-representation in pollen diagrams (van Zeist et al., 1975; Fægri and Iversen, 1989). Pollen related to grazing or disturbed areas like *Plantago lanceolata* and *Rumex* are very scarce. Probably pastureland was reduced while agriculture seems to have played an important role in the economy. Marinova

et al. (2012) pointed out that after a period with no archaeological evidence, a remarkable socio-economical expansion is attested since the 6th century BC, at around 2500 yr BP.

Either a climate change or a large-scale human impact can be invoked to explain the strong reduction of *Pinus* and *Abies* occurred during D-9 after 2000 yr BP. First Greek, followed by Romans and Byzantines could have regionally cut them for valuable characteristics of the timber. Macedonia was indeed important region for timber harvesting since the 5th century AD

and probably earlier. Fir and pine were the standard ship-timber and naval power depended on them (Harris, 2013). Their pronounced decrease is again related with the presence of *Juniperus* that typically takes advantage in more degraded environments. At the same time, both concentration and influx reach very low values, mirroring decreasing PAHs. Cereals, *Castanea*, *Juglans* and *Olea* cultivation together with ruderal and weed taxa attest a strong human impact. Historical reconstructions confirm that in Macedonia the culmination of the growth in both cereals and walnut occurred during the Mid-Roman and Late Antique

periods (1650-1500 yr BP; Izdebski et al., 2015).

A strong reduction of AP % (Fig. 5), caused by a decrease of oaks or pines, is found at many sites around 2000 BP. In particular, a *Pinus* drop is found at Prespa at 2300 yr BP (Panagiotopoulos et al., 2013), at Trilistnika at 2000 yr BP (Tonkov et al., 2008) and probably also at Eski Acigöl (Woldring and Bottema, 2003) even if for the Turkish site the chronology remains uncertain. In all sites this change marks forest reduction. It can once more been interpreted either as a transition from wetter to dryer conditions or as a strong human-induced forest clearance. Climatic data are available from tree ring-based reconstructions of Central Europe (Büntgen al., 2011). The authors rule out a decrease in total reconstructed precipitation and temperature anomalies at 2100 yr BP. A global climate change seems, however, to be ruled out for this period (Bond et al., 1997).

The new spread of *Pinus* occurring in D-10 might not indicate a recover of the forest as mesophilous trees percentages are decreasing. Pines can in fact have a pioneer role in anthropogenic-influenced landscapes (Litt et al., 2012), as they quickly grow in degraded areas such as those colonized by junipers. At ca. 700 yr BP, high pollen concentration and influx values match high TOC, increased polycyclic aromatic hydrocarbon and fecal stanol concentrations and slightly increased Cichorioideae that could be related to pastoralism (Florenzano et al., 2015). Although decreased temperatures are detected in geochemical proxies, Little Ice Age (LIA) is not evident in pollen data. LIA could be seen in the forest recover indicated by increasing influx values (Fig. 3) but not in the pollen percentage assemblage that still remains high. The strong human activities consisting in livestock farming, fire use, cultivation, may mask the climate signal and overlap the natural changes of LIA. A strong land-use is still present in the top samples, with hints of *Olea*, *Juglans* and *Vitis* cultivation in the last few centuries.The expansion of these taxa corresponds to the economic and demographic one reported by (Gogou et al., 2016) at around 900 yr BP as a consequence of new political order, when Macedonian region becomes of central importance for the Byzantine Empire. *Olea* and *Vitis* cultivation has been and still is of primary importance in the Byzantine diet (Xoplaki et al., 2016). Nowadays grape is the primary product of Dojran area, even if today there are few farms growing *Vitis vinifera* on altogether 263 ha (Bojovic et al., 2016). The increase of Poaceae recorded in D-10 can be ascribed to the presence of a littoral belt. Modern vegetation suggests the presence of either *Paspalum distichum* (water finger-grass) or *Phragmites australis* (common reed). Common reeds have been recently used in the area for different purposes, primarily to produce special traps for fishing (Bojovic et al., 2016).

## 6   Conclusions

The new high-resolution pollen record from Lake Dojran draws the vegetation history of the area between FYROM and Greece for the last 12500 years evidencing the regional response to climate forcing. The pollen record covers the environmental changes from the Late Glacial until today passing from a natural undisturbed landscape to one when increasing anthropogenic influences overlap climate change. These new data, together with sedimentological, biomarker and diatom data available from the same core, highly contribute to better understand the environmental history, including both climatic and human evidences. A steppe characterized by Amaranthaceae, *Artemisia* and Poaceae prevails during Late Glacial period. The beginning of the Holocene reforestation at Lake Dojran is dated at 11500 yr BP, similar to other mediterranean records. It consists of a false start of AP found also in other sites, like Orestiás, Lake Prespa, Ioannina and Lake Pergusa, and preceding the real expansion of the Holocene forest. Two millennia were necessary to reach maturity in the forests dominated by deciduous *Quercus*. Mesophilous

vegetation dominated for the entire Holocene, but a well-developed forest is found until ca. 4000 yr BP. First human traces are recorded around 5000 yr BP with the presence of cereals, *Juglans* and *Rumex*. Synanthropic taxa rise considerably during Bronze Age and particularly in the Roman age. Around 2000 yr BP, the arboreal pollen and mainly *Pinus* show a strong reduction that is detectable at most sites of the mediterranean basin. Even if the attestation of drier conditions can be advocated to explain the abrupt change in vegetation, the forest clearance made by the Roman Empire would have been so strong to affect vegetation on regional scale. Present day vegetation is human-influenced, but the natural vegetation is still preserved, confirming the importance of the region as a biodiversity conservation area.

The comparison of the Dojran pollen record with the regional ones evidences similarities in the reafforestation dynamics at the beginning of the Holocene. The detailed sequence clarifies and better describes some peculiarities like the characteristic interruption in the afforestation at the beginning of the Holocene (between ca. 10800 and 11500 yr BP). Although this behaviour was present in other proxies, the Dojran sequence clearly evidences this dynamic allowing identifying the same behaviour in the region. The high arboreal cover that characterized the mid-Holocene is more evident at Dojran where the fluctuation in the arboreal pollen percentage are minimal despite of the high analitical detail. Finally, the clear AP decrease, related to the strong reduction of pine since 2200 yr BP, is now detectable in the regional proxies where, since now, it has been ever linked. The presented data represent the first Late Glacial and Holocene continuous sequence of Lake Dojran. The high-resolution pollen analysis and the available multi-proxy dataset provide a unique contribution to the reconstruction of vegetation dynamics and their relationship with palaeoenvironmental changes and human impact of the entire region.

*Data availability.* All the pollen data published in the present paper is available through the online database Pangaea at https://doi.pangaea.de/10.1594/PANGAEA.885797 (Masi et al., 2017). All the geochemical data already published by Francke et al. (2013a) are available through the same database (https://doi.org/10.1594/PANGAEA.860791; Francke et al., 2013b). Biomarkers shown in figure 4 and published by Thienemann et al. (2017) are available through the same database at https://doi.pangaea.de/10.1594/PANGAEA.885797 (Masi et al., 2017).

*Author contributions.* The manuscript was written by A. Masi (all paragraphs) and L. Sadori (paragraphs 2, 4, 5, 6) with the substantial contribution of B. Wagner (paragraphs 1, 5, 6), A. Francke (paragraphs 4, 5, 6) and M. Thienemann (paragraphs 4, 5, 6). Pollen analysis was carried out by A. Masi with the contribution of C. Pepe for the chemical procedure and microscope identification. A. Masi is responsible for data management, elaboration of figures and diagrams.

*Competing interests.* The authors declare that they have no conflict of interest.

*Acknowledgements.* We would like to thank all the authors of the papers on sedimentological and stable isotope (Francke et al., 2013a), diatom (Zhang et al., 2014), and biomarker data (Thienemann et al., 2017) of Lake Dojran that have been precious in the interpretation of pollen data. We are grateful to Katerina Kouli and Kostantinos Panagiotopoulos for providing pollen data of Lake Orestiás and Prespa. We express gratitude to prof. Vlado Matevski for the support in the description of modern vegetation of the Dojran area.

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

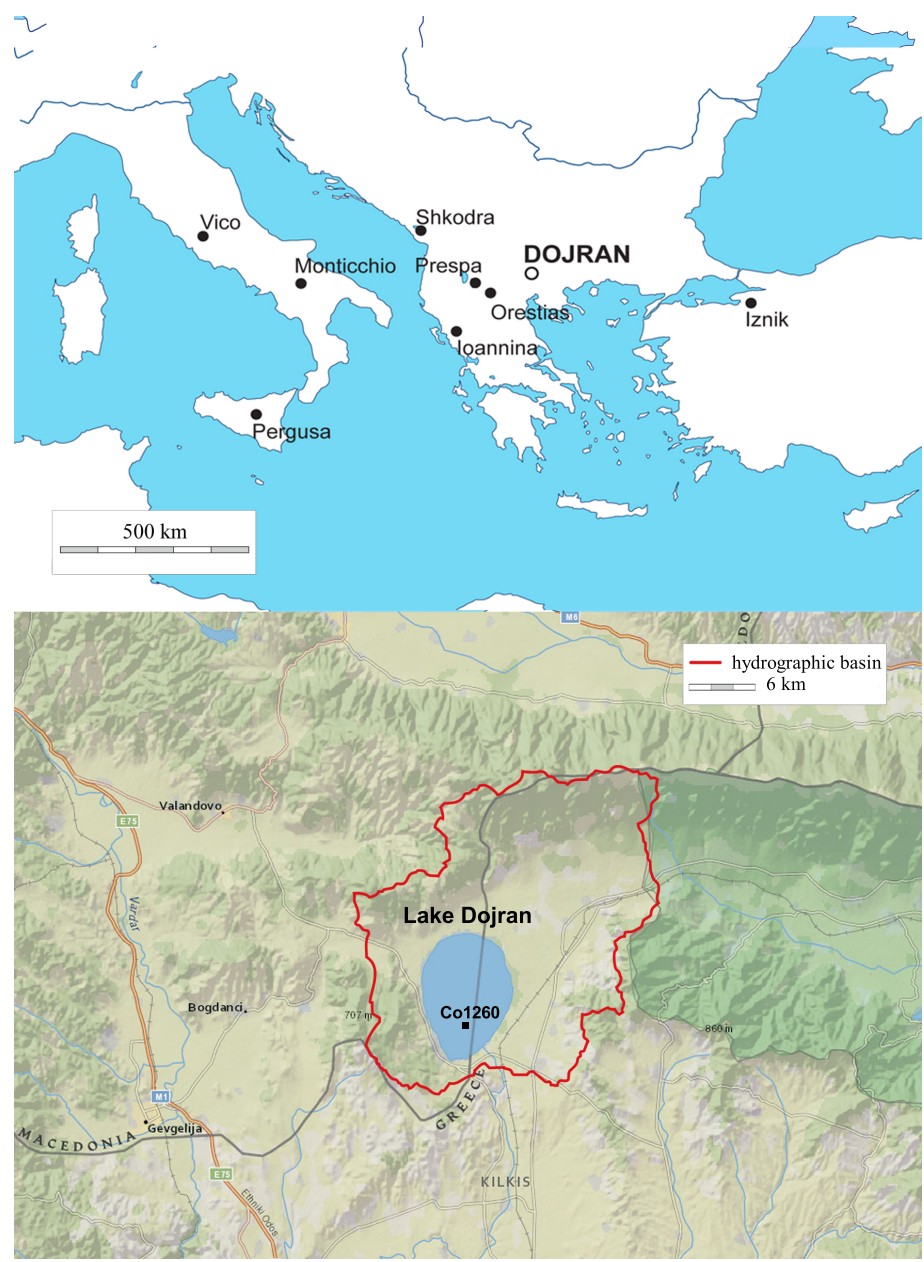

**Figure 1.** Top: Location of Lake Dojran and of the other sites mentioned in the paper. Bottom: detail of the lake and its surrounding, the hydrological basin and the location of the analysed core Co1260. The map is taken from Basemap Esri.

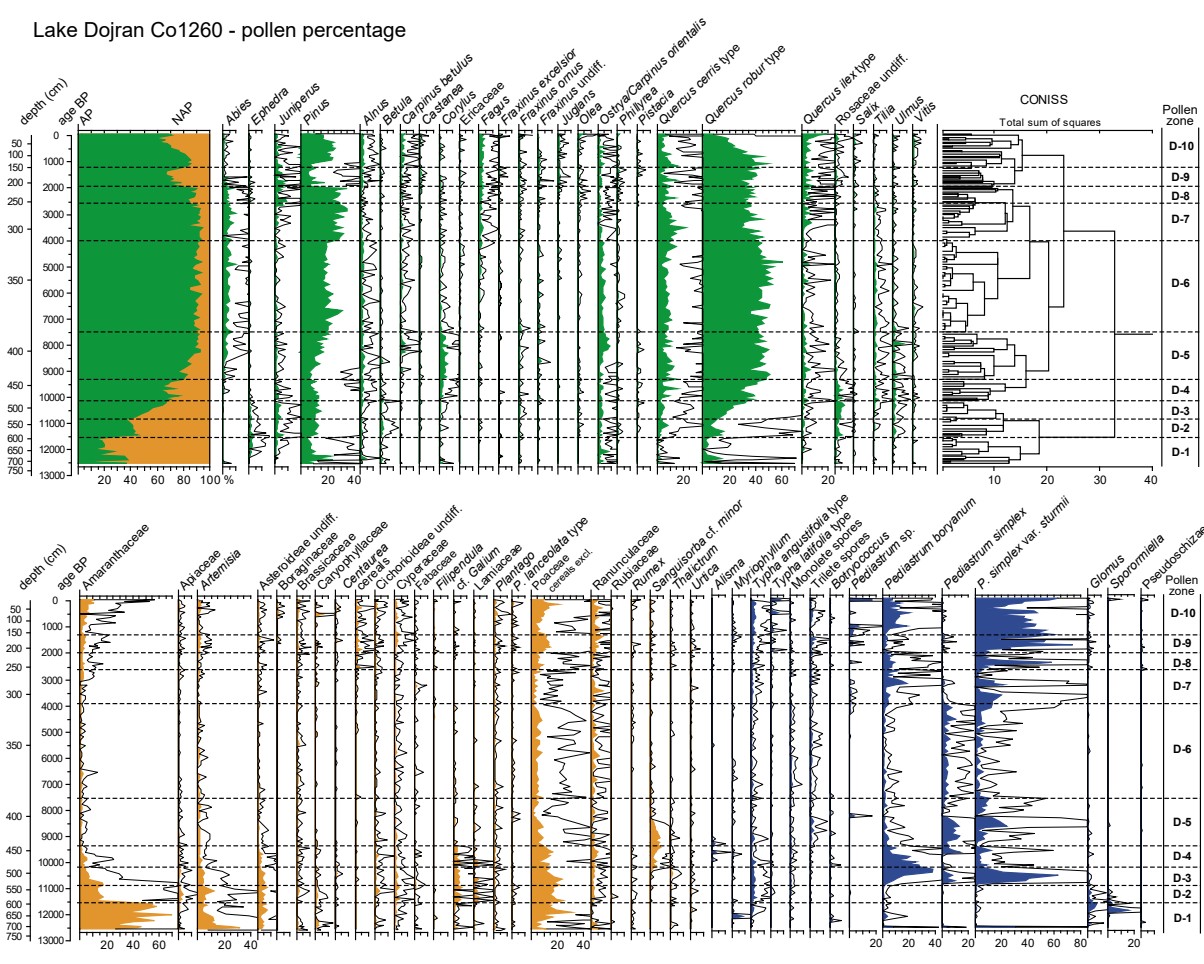

**Figure 2.** Pollen percentage diagrams of arboreal, non arboreal and NPP selected taxa and CONISS. Curve magnification 5x.

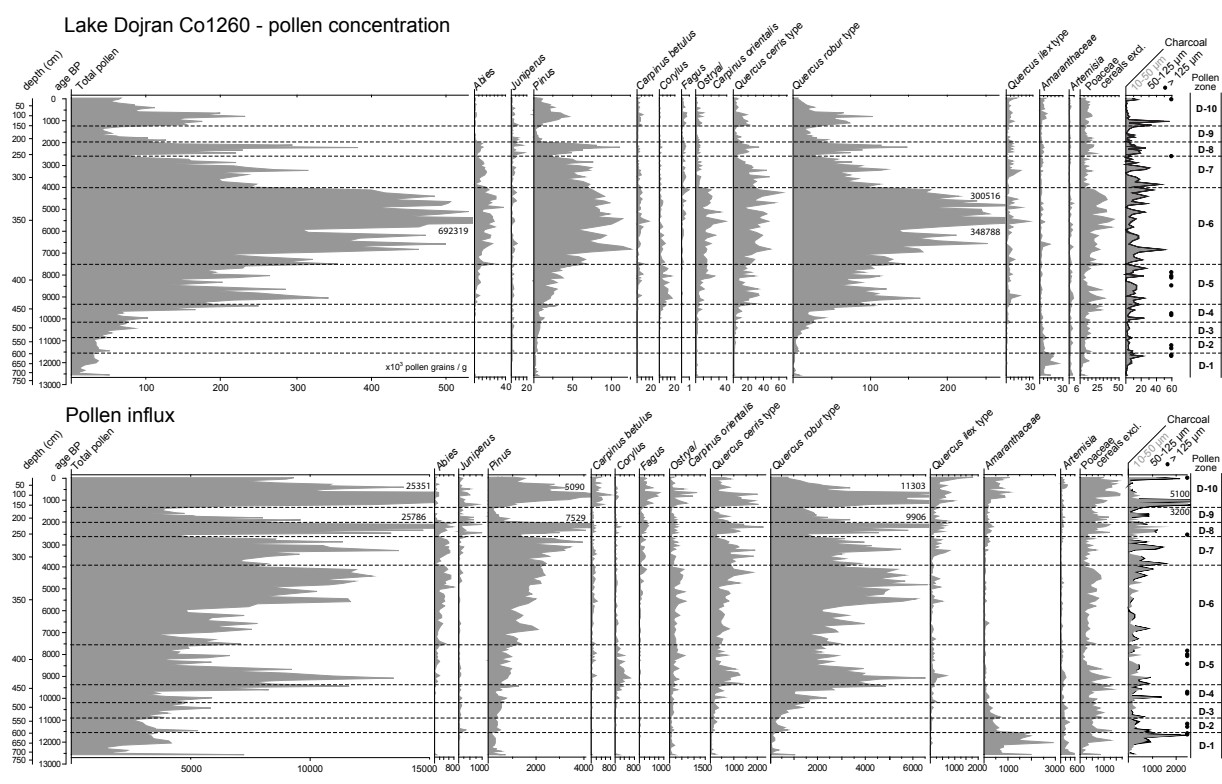

**Figure 3.** Concentration (top) and influx (bottom) pollen diagrams of selected taxa and charcoals.

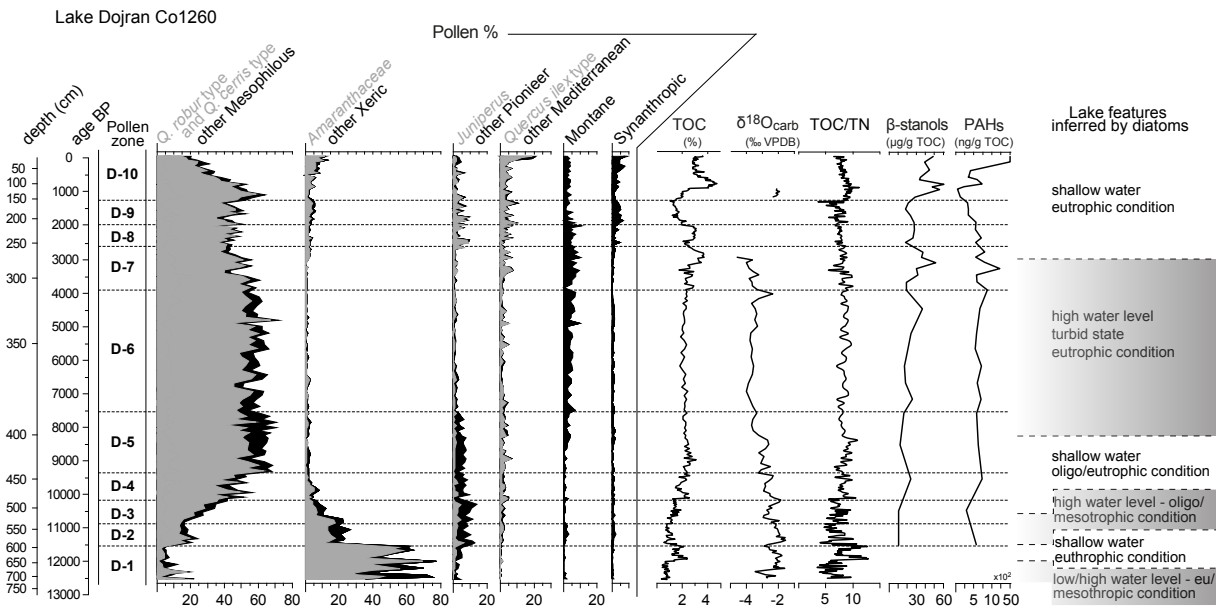

**Figure 4.** Pollen percentage diagrams of plant groups. Mesophilous: *Acer*, *Buxus*, *Carpinus betulus*, *Celtis*, *Corylus*, *Fraxinus excelsior/oxycarpa*, *Hedera*, *Ostrya/Carpinus orientalis*, *Quercus robur* type, *Quercus cerris* type, *Tilia*, *Ulmus*. Xeric: Amaranthaceae, *Artemisia*, *Helianthemum*. Pioneer: *Betula*, *Corylus*, *Ephedra*, *Juniperus*, Rosaceae. Mediterranean: *Arbutus*, *Fraxinus ornus*, *Cistus*, *Pistacia*, *Phillyrea*, *Quercus ilex* type, *Rhamnus*. Montane: *Abies*, *Betula*, *Fagus*. Synanthropic: *Castanea*, *Centaurea cyanus*, cereals (*Secale, Triticum, Hordeum*), *Juglans*, *Olea*, *Plantago lanceolata*, *Rumex*, *Trifolium*, *Vitis*. Comparison of pollen data with selected geochemical data (Francke et al., 2013a), biomarker curves (Thienemann et al., 2017) and lake features inferred by diatom data (Zhang et al., 2014) of Lake Dojran.

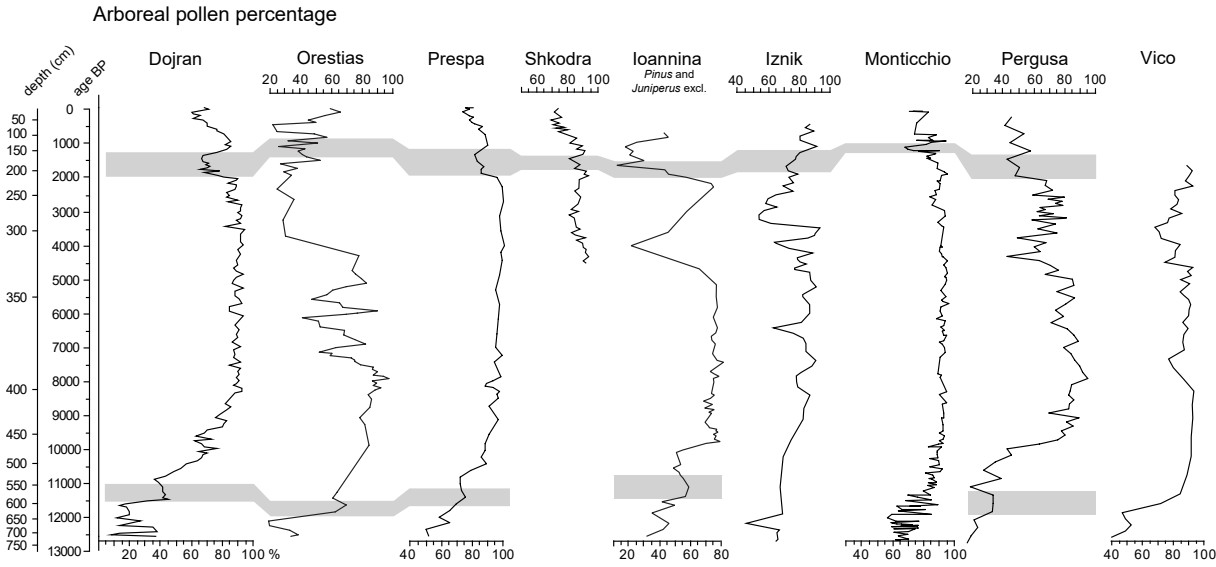

**Figure 5.** Arboreal pollen percentage diagram from Lake Dojran, Lake Prespa, Ioannina basin, Lake Pergusa, Lake Iznik, Lago Grande di Monticchio, Lake Vico, Orestiás and Lake Shkodra. See text for references.