# Peer review of "Vegetation history and palaeoclimate at Lake Dojran (FYROM/Greece) during the Late Glacial and Holocene"

_Climate of the Past, 2017_

## Referee Comment (RC1) · A.M. Mercuri (Referee) · 16 Oct 2017

This is a high resolution and important paper adding new evidence of vegetation changes under climate and human actions in the eastern Mediterranean. Pollen data are precise and detailed, and the interdisciplinary research has high potentiality to deepen the palaeoenvironmental changes of the region. The complexity of results is largely described in the relevant section. The discussion needs some further comments by avoiding loss of information or partial consideration of the interlaced nature-culture dynamics involved here as in other Mediterranean sites. The pollen diagram is exceptionally useful to follow climate oscillations in the region, to understand the vegetation dynamic of a conservative site, and also provide some interesting information on the presence of past cultures in the area. The paper is highly recommended for publication on Climate of the past, after addressing the following points:

Abstract: it is well written and interesting, I suggest however to strenghen the importance of the new data obtained from this high resolution and multidisciplinary study. The last sentence "The Lake Dojran multi-proxy analysis including pollen data provide a valuable contribution to the palaeoenvironmental reconstruction and the comprehension of the past vegetation dynamics of southern Balkans." is quite general and vague while it should be more useful to add some concrete details on the contribution.

Pag. 2, line 1 and 5 = "understanding past climate and its evolution" : this sentence is not linked to the next paragraph where you introduce 'vegetation dynamics'; I suggest to change the first sentence as "understanding past climate changes and vegetation dynamics".

Pag. 3, line 7 = smallest Line 27-31= put latin names in italics; line 34 = 'sylvestris'

Pag- 4, line 20 = delete 'only'; line 29 = please, specify the main pollen sum for percentage calculations (all pollen? total land pollen?) Pag 5, lines 15-25 = check, some parts seem methods rather than results

D1 = gradual decrease of Artemisia and increase of Quercus = LateGlacial phases; fragmented Pinus = alluvial deposits?

Pag. 6, line 32 = Pinus 15% does not indicate a local grow of pine trees

Pag. 7, zone D4 = there is the disappearance of Ephedra, decrease of Artemisia, end of Centaurea, presence of Alisma (and sensible decrease of Betula); Galium and Filipendula has the last high values and then will decrease in the next zone = most pollen evidence points to a wet phase, but from your text it seems that this relies on sedimentological data ("According to the sedimentological data, this time period was characterized by increasing humidity in particular during summer.")

Pag. 7, line 28-29 = "The rising AP% in zone D-3 to D-5 is paralleled by a decreasing trend in the average chain length of vascular plant n-alkanes, also indicating increasing arboreal vegetation" = this sentence is not useful here or should be reformulate/completed: practically, you write that 'rising AP' indicates 'increasing arboreal vegetation' and refer to the n-alkanes curve without figure citation (and the reference to published paper is some pages before). I think that here (in the result section) you should point to the increase of Quercus, Abies, Pinus which give a great contribution to this gradual increase matching the rise of wet (and cool?) conditions. In general, in the result section, I suggest to point to your palynological data to obtain palaeoclimatic /palaeoecological inferences because data are strong and clear.

Pag. 8, line 9 = cereal traces are present even before; you have 'Cerealia type' pollen grains that may belong to wild species with large pollen – you should mention this in the method section, and therefore probably the term 'Cerealia type' may be more appropriate to this case than others

Pag. 8, line 24 =" Cereals (>0 %)" ?

Pag. 9 Vitis was high even in D6 OJC+Vitis? = I do remark that the OJC group (Fig. 2) shows a clear trend in your diagram (if Olea+Juglans+Castanea are summed up, the curve has a clear increase at around 2500 BP, to which they contribute in this order: first Olea, then Juglans then Castanea) while the scattered curve of Vitis has not the same sharp difference between the 'before' and 'after' the 2500 BP. The Vitis curve is not comparable and has not the same significance of OJC: I cannot agree with your sentence "These four taxa show slightly different behaviours and are sporadically present since the early Holocene." because it is not what we see in the diagram

Pag. 10 line22 = "Fluctuating presence of coniferous and deciduous taxa in glacial periods is recorded in several lakes" : Fluctuating presence of (coniferous and deciduous) taxa is recorded during the Late Glacial oscillations. Don't you have a Bolling/Allerod – Younger Dryas wet-warm/dry-cold oscillation in your zone D1? Also, of interest,

is that you have a sharp increase of AP curve (D2) but then (D3) each tree (Quercus, Rosaceae, Ulmus, Juniperus,...) has a gradually increasing curve meaning that this is truly a vegetational (rather than floristic) reply. This gradual increase matches the gradual decline of the steppe of Artemisia+Am/Chenopodiaceae showing a fairly conservative-resilient ecosystem that characterises this area and many lakes you cited.

Pag 12 lines 1-3 = as you mention Bronze Age, and Neolithic, please, put the relevant millennia in brackets to show the chronology of these phases

"The introduction of Juglans in the Balkans is usually dated at ca. 3000 yr BP (Sadori et al., 2013)." = do you mean that the earlier pollen record is found at c. 3000 BP in another lake? Please, explain and then: "In this frame, the early presence of walnut "... in your Dorjan record?

Line 14: "In central Italy, a decrease in humidity, detected soon before 4000 yr BP, is found in low-stand lake levels (Giraudi et al., 2011) and in speleothems (Zanchetta et al., 2016)." Possibly this could be more related to the trends of some pollen curve of your record. I noted that your diagram shows a very similar trend to what I find in the Adriatic core RF93-30: The fall of Quercus ilex type occurred in core RF93-30 at around 4130 cal. B.P., and it is contemporary with both the rise of deciduous oaks (your Q.cerris, I mean) and the thinning of Abies and Juniperus type (this latter is less evident in your record) " (Mercuri et al. 2012, p. 362). The dryness trend in the marine record, however, also caused a gradual decrease of Fagus

Line 26: I suggest to revise the consideration of Vitis as unambiguously included in the 'cultivated taxa' - look at your Vitis curve

Line 27 = "Mercuri et al. (2013) introduced the OJC (Olea, Juglans, Castanea) sum to estimate the rate of human impact on natural ecosystems" – the sum was firstly calculated to follow the development of cultural landscape in central Mediterranean (see also Mercuri 2014 Landscape Ecology, and also for the on site/off site record interpretation). The human impact was especially investigated by the API sum (Anthropogenic

Pollen indicators; Mercuri et al. 2013 Annali di Botanica) which calculate the percentage sum of the seven pollen typologies found in archaeological site layers: the API sum indicates the flora which is common (ubiquitarian) near the sites where humans lived.

Line 19 = are your pollen grains Cichorioideae or Cichorieae ? according to the paper by Florenzano et al 2015, only Cichorieae have fenestrate pollen

Line 22 = You mention that "The strong human activities consisting in livestock farming, fire use, cultivation, overlapped the natural changes of LIA. " but the trends of your pollen curves are not all evidences (again, and also) that there was an impressive resilience in vegetation dynamics of this area?

Line 28 = this is the second time you refer to Paspalum and Phragmites: I know that it is possible to identify Phragmites, but is it also reliable to identify Paspalum (put it in the methods)? If you, however, are referring to the modern vegetation you know from vegetation papers, please specify and change a bit the sentence

Pag 13, Line 6 = "found also in other sites": mention these sites or delete these words, as you have just said the same thing in the previous sentence. In general, you said several times that you compare data but there are sites that are present only in the figure and not mentioned in the text. A general comment referring to the figure 5 – also outlining the remarkable similarities or dissimilarities – should be added as a conclusive consideration of peculiarities and relevance of Dojrian. The different elements visible in the diagram should be distinguished in the final sentences, describing the importance of this off-site record. Also, I suggest incorporating in the conclusions a comment on gradual trends, and resilience

Minor typos 'Mediterranean' with capital letters (?) Legend of Fig. 4 = Buxusù

---

## Referee Comment (RC2) · Anonymous Referee #2 · 18 Oct 2017

This paper is an important addition to the study of the vegetation history and paleoclimatology of the Balkans in the Holocene. It presents new pollen data and compares to other proxies produced by some of the authors and published previously. The authors' main findings are that there was a rise in arboreal vegetation at the beginning of the Holocene but that it takes a couple of millennia to be definitively attested. This supports the findings of many previous studies from the Mediterranean region. The fact most taxa are present at the bottom of the core suggests a glacial refugia in the catchment. They also find very interesting first signs of human presence ∼5ka (cereal pollen) and then various human impacts through the next few millennia. The methodology and interpretation is robust, there are new findings that are of interest to the community, and

it is generally well written with good quality figures, so I recommend this for publication if the following points are considered (especially in the discussion).

While I feel the interpretations are sound and the stories interesting, the authors could expand on the work by comparing their results and findings more to other studies and thinking about the wider implications and underlying climate dynamics that may have caused the observed changes. E.g.:

1. The authors do compare their results to those from further afield e.g. Italy and Greece. However, there are very similar patterns of slow arboreal pollen increase even further afield e.g. in central and eastern Turkey (Van, Eski) too – may be worth discussing this and how all across the Balkans and eastern Mediterranean this slow increase is being picked up, and what this tells us about what was happening with climate/vegetation change.

2. More discussion of why there is no response in the pollen record to the 8.2ka event would be good – you talk about the lack of response in the pollen record, which is really interesting – but it would be nice to hear some thought on why there was no response, when there was in e.g. Tenaghi Philippon's pollen record (Pross et al., 2009).

3. "Athanasiadis et al. 2000 provided pollen results for littoral cores from Lake Dojran covering the last 5000 years" – could you compare your pollen record to theirs? Are there differences?

4. In the results section 4.3, you talk about how several different proxies indicate increasing humidity in this pollen zone, but the diatom assemblage data suggest relatively shallow waters – it would be nice to offer some reasons for this discrepancy (even if these are discussed in the Zhang et al. paper already).

Minor points: "Yugoslav" not "Yugoslavian" – page 1, line 2. Nutrients rather than nutrient on page 1, line 12? Check capitalization of 'Mediterranean' throughout.

---

## Short Comment (SC1) · 28 Oct 2017

The PAGES Data Stewardship Integrative Activity seeks to advance best practices for sharing the data generated and assembled as part of all PAGES-related activities. The CP Special Issue, "PAGES Young Scientists Meeting 2017" is part of this PAGES activity. The co-editors of the Special Issue are reviewing the data availability within each of the CP-Discussion papers in relation to the CP data policy (https://www.climate-of-the-past.net/about/data_policy.html) and current best practices. The editor team is making recommendations for each paper, with the goal of achieving a high and consistent level of data stewardship across the Special Issue. We recognize that an additional effort

will likely be required to meet the high level of data stewardship envisaged, and we appreciate the dedication and contribution of the authors. This includes the use of Data Citations (see example below). Authors are also strongly encouraged to deposit significant code into a suitable repository and to cite it using a Data Citation.

We ask authors to respond to our comments as part of the regular open interactive discussion. If you have any questions about PAGES Data Stewardship principles, please contact any of us directly. Best wishes for the success of your paper.

YSM Special Issue editor team

R. Barnett, D.S. Kaufman, M.F. Loutre, M.N. Evans, S.C. Fritz, C. Tabor, H. Plumpton, Y. Zhang, E. Razanatsoa, and E. Dearing Crampton Flood

For this paper: (1) Research input data: geochemical proxies, diatoms, biomarkers

This research contribution discusses widely the interrelationship between published proxy data (geochemistry, diatoms, biomarkers) from Francke et al. (2013), Zhang et al. (2014) and Thienemann et al. (2017) and new pollen data (this study) from Lake Dojran. In order to adhere to the Data Sharing Policy for submissions to Climate of the Past, we request that the authors: (a) work with the authors of Francke et al. (2013), Zhang et al. (2014) and Thienemann et al. (2017) to submit the published geochemistry, diatom and biomarker data to a long-standing online data repository (the data shown in Fig. 4), and; (b) obtain and provide a Data Citation or URL link for access to these data. If the data currently exist in a repository, then only part (b) is necessary.

(2) Research output data: pollen

This research contribution presents abundant new and valuable pollen data from Lake Dojran. In order to adhere to the Data Sharing Policy for submissions to Climate of the Past, we request that the authors submit these new data to a long-standing online data repository, and obtain and provide a Data Citation or URL link for access to these

data. The statement "pollen dataset may be accessed by request to authors" is not an acceptable alternative.

————-

What is a "Data Citation"? Data Citations track the provenance of a dataset giving credit to the data generator; this is in addition to any references to publications where the data are described. Data Citations are used in the text (or tables) alongside and in the same way as publication citations. In the Reference list, they include: Creators, Title, Repository, Identifier, Submission Year. More information about Data Citations is here: <https://www.datacite.org/mission.html> Here is an example of text and corresponding citations (using CP punctuation style):

"The PAGES2k Consortium (2017a) assembled a large global dataset of temperature-sensitive proxy records (PAGES2k Consortium, 2017b). Among the records is the paleo-temperature reconstruction from Laguna Chepical (de Jong et al., 2016), which was described by de Jong et al. (2013)."

References

de Jong, R., von Gunten, l., Maldonado, A., and Grosjean, M.: Late Holocene summer temperatures in the central Andes reconstructed from the sediments of high-elevation Laguna Chepical, Chile (32° S), Climate of the Past, 9, 1921-1932, 2013.

de Jong, R., von Gunten, l., Maldonado, A., and Grosjean, M.: Laguna Chepical summer temperature reconstruction, World Data Center for Paleoclimatology, https://www.ncdc.noaa.gov/paleo/study/20366, 2016.

PAGES 2k Consortium: A global multiproxy database for temperature reconstructions of the Common Era, Scientific Data, 4,170088, 2017a.

PAGES 2k Consortium: A global multiproxy database for temperature reconstructions of the Common Era, version 2.0.0, figshare, https://figshare.com/s/d327a0367bb908a4c4f2, 2017b.

---

## Author Comment (AC1) · 4 Dec 2017

We want to thank Referee #1 for her careful and constructive review of our paper. We will take into account her valuable advice for the revision of the manuscript.

Reviewer: Abstract: it is well written and interesting, I suggest however to strenghen the importance of the new data obtained from this high resolution and multidisciplinary study. The last sentence "The Lake Dojran multi-proxy analysis including pollen data provide a valuable contribution to the palaeoenvironmental reconstruction and the comprehension of the past vegetation dynamics of southern Balkans." is quite general and vague while it should be more useful to add some concrete details on the contribution.

[Figure]

Authors: THE REFEREE IS RIGHT, THE ABSTRACT WILL BE CHANGED ACCORD-
INGLY

R: Pag. 2, line 1 and 5 = "understanding past climate and its evolution" : this sentence
is not linked to the next paragraph where you introduce 'vegetation dynamics'; I suggest
to change the first sentence as "understanding past climate changes and vegetation
dynamics".

A: WE AGREE, THE REQUESTED CHANGES WILL BE DONE

R: Pag. 3, line 7 = smallest Line 27-31= put latin names in italics; line 34 = 'sylvestris'
Pag- 4, line 20 = delete 'only'; line 29 = please, specify the main pollen sum for per-
centage calculations (all pollen? total land pollen?) Pag 5, lines 15-25 = check, some
parts seem methods rather than results

A: WE AGREE, THE REQUESTED CHANGES WILL BE DONE

R: D1 = gradual decrease of Artemisia and increase of Quercus = LateGlacial phases;
fragmented Pinus = alluvial deposits?

A: AUTHORS AGREE WITH REVIEWER'S INTERPRETATION THAT WILL BE RE-
PORTED IN THE DISCUSSION SECTION. IN THE RESULTS WE LIMITED AS MORE
AS POSSIBLE THE INTERPRETATION AND PRESENTED ONLY THE RESULTS

R: Pag. 6, line 32 = Pinus 15% does not indicate a local grow of pine trees

A: WE AGREE, THE REQUESTED CHANGES WILL BE DONE

R: Pag. 7, zone D4 = there is the disappearance of Ephedra, decrease of Artemisia,
end of Centaurea, presence of Alisma (and sensible decrease of Betula); Galium and
Filipendula has the last high values and then will decrease in the next zone = most
pollen evidence points to a wet phase, but from your text it seems that this relies on
sedimentological data ("According to the sedimentological data, this time period was
characterized by increasing humidity in particular during summer.")

A: WE AGREE, THE REQUESTED CHANGES WILL BE DONE

R: Pag. 7, line 28-29 = "The rising AP% in zone D-3 to D-5 is paralleled by a decreasing trend in the average chain length of vascular plant n-alkanes, also indicating in- creasing arboreal vegetation" = this sentence is not useful here or should be reformulate/completed: practically, you write that 'rising AP' indicates 'increasing arboreal vegetation' and refer to the n-alkanes curve without figure citation (and the reference to published paper is some pages before). I think that here (in the result section) you should point to the increase of Quercus, Abies, Pinus which give a great contribution to this gradual increase matching the rise of wet (and cool?) conditions. In general, in the result section, I suggest to point to your palynological data to obtain palaeoclimatic /palaeoecological inferences because data are strong and clear.

A: THE SENTENCE WILL BE MOVED

R: Pag. 8, line 9 = cereal traces are present even before; you have 'Cerealia type' pollen grains that may belong to wild species with large pollen – you should mention this in the method section, and therefore probably the term 'Cerealia type' may be more appropriate to this case than others.

A: WE AGREE, THE REQUESTED CHANGES WILL BE DONE

R: Pag. 8, line 24 =" Cereals (>0 %)" ?

A: THE TYPO WILL BE CORRECTED

R: Pag. 9 Vitis was high even in D6 OJC+Vitis?  = I do remark that the OJC group (Fig. 2) shows a clear trend in your diagram (if Olea+Juglans+Castanea are summed up, the curve has a clear increase at around 2500 BP, to which they contribute in this order: first Olea, then Juglans then Castanea) while the scattered curve of Vitis has not the same sharp difference between the 'before' and 'after' the 2500 BP. The Vitis curve is not comparable and has not the same significance of OJC: I cannot agree with your sentence "These four taxa show slightly different behaviours and are sporadically

present since the early Holocene." because it is not what we see in the diagram

A: WE AGREE, THE REQUESTED CHANGES WILL BE DONE

R: Pag. 10 line22 = "Fluctuating presence of coniferous and deciduous taxa in glacial periods is recorded in several lakes" : Fluctuating presence of (coniferous and deciduous) taxa is recorded during the Late Glacial oscillations. Don't you have a Bolling/Allerod – Younger Dryas wet-warm/dry-cold oscillation in your zone D1? Also, of interest, is that you have a sharp increase of AP curve (D2) but then (D3) each tree (Quercus, Rosaceae, Ulmus, Juniperus,. . .) has a gradually increasing curve meaning that this is truly a vegetational (rather than floristic) reply. This gradual increase matches the gradual decline of the steppe of Artemisia+Am/Chenopodiaceae showing a fairly conservative-resilient ecosystem that characterises this area and many lakes you cited.

A: WE WILL USE THE REFEREE SUGGESTION TO ENRICH THE DISCUSSION

R: Pag 12 lines 1-3 = as you mention Bronze Age, and Neolithic, please, put the relevant millennia in brackets to show the chronology of these phases

A: AUTHORS WILL ADD THE INFORMATION WHEN AVAILABLE IN THE ORIGINAL ARTICLES. UNFORTUNATELY MANY ARCAHEOLOGAL PAPERS USUALLY AVOID TO BE CLEAR IN THE CHRONOLOGICAL FRAMING

R: "The introduction of Juglans in the Balkans is usually dated at ca. 3000 yr BP (Sadori et al., 2013)." = do you mean that the earlier pollen record is found at c. 3000 BP in another lake? Please, explain and then: "In this frame, the early presence of walnut ". . . in your Dorjan record?

A: TEXT WILL BE WRITTEN MORE CLEARLY

R: Line 14: "In central Italy, a decrease in humidity, detected soon before 4000 yr BP, is found in low-stand lake levels (Giraudi et al., 2011) and in speleothems (Zanchetta et al., 2016)." Possibly this could be more related to the trends of some pollen curve

of your record. I noted that your diagram shows a very similar trend to what I find in the Adriatic core RF93-30: The fall of Quercus ilex type occurred in core RF93-30 at around 4130 cal. B.P., and it is contemporary with both the rise of deciduous oaks (your Q.cerris, I mean) and the thinning of Abies and Juniperus type (this latter is less evident in your record) " (Mercuri et al. 2012, p. 362). The dryness trend in the marine record, however, also caused a gradual decrease of Fagus

A: A: WE WILL USE THE REFEREE SUGGESTION TO ENRICH THE DISCUSSION

R: Line 26: I suggest to revise the consideration of Vitis as unambiguously included in the 'cultivated taxa' - look at your Vitis curve

A: TEXT WILL BE CHANGED

R: Line 27 = "Mercuri et al. (2013) introduced the OJC (Olea, Juglans, Castanea) sum to estimate the rate of human impact on natural ecosystems" – the sum was firstly calculated to follow the development of cultural landscape in central Mediterranean (see also Mercuri 2014 Landscape Ecology, and also for the on site/off site record interpretation). The human impact was especially investigated by the API sum (Anthropogenic Pollen indicators; Mercuri et al. 2013 Annali di Botanica) which calculate the percentage sum of the seven pollen typologies found in archaeological site layers: the API sum indicates the flora which is common (ubiquitarian) near the sites where humans lived.

A: TEXT WILL BE CHANGED ACCORDING TO REFEREE'S INDICATION

R: Line 19 = are your pollen grains Cichorioideae or Cichorieae ? according to the paper by Florenzano et al 2015, only Cichorieae have fenestrate pollen

A: YES, THEY ARE CICHORIEAE. WE INCREASE THE DETAIL OF OUR POLLEN IDENTIFICATION THANKS TO THE REFEREE'S INDICATION. THE TEXT HAS BEEN CHANGED

R: Line 22 = You mention that "The strong human activities consisting in livestock

farming, fire use, cultivation, overlapped the natural changes of LIA. " but the trends of your pollen curves are not all evidences (again, and also) that there was an impressive resilience in vegetation dynamics of this area?

A: TEXT WILL BE CHANGED

R: Line 28 = this is the second time you refer to Paspalum and Phragmites: I know that it is possible to identify Phragmites, but is it also reliable to identify Paspalum (put it in the methods)? If you, however, are referring to the modern vegetation you know from vegetation papers, please specify and change a bit the sentence

A: YES, WE REFER TO MODERN VEGETATION. THE SENTENCE WILL BE CHANGED

R: Pag 13, Line 6 = "found also in other sites": mention these sites or delete these words, as you have just said the same thing in the previous sentence. In general, you said several times that you compare data but there are sites that are present only in the figure and not mentioned in the text. A general comment referring to the figure 5 – also outlining the remarkable similarities or dissimilarities – should be added as a conclusive consideration of peculiarities and relevance of Dojran. The different elements visible in the diagram should be distinguished in the final sentences, describing the importance of this off-site record. Also, I suggest incorporating in the conclusions a comment on gradual trends, and resilience

A: WE FOLLOW THE REFEREE SUGGESTION IMPROVING THE TEXT

---

## Author Response (AR2)

RESPONSE TO REVIEWER 1

Authors: WE THANK THE REFEREE #1 FOR USEFUL AND CONSTRUCTIVE COMMENTS. PLEASE FIND OUR REPLIES BELOW.

R: Abstract: it is well written and interesting, I suggest however to strenghen the importance of the new data obtained from this high resolution and multidisciplinary study. The last sentence "The Lake Dojran multi-proxy analysis including pollen data provide a valuable contribution to the palaeoenvironmental reconstruction and the comprehension of the past vegetation dynamics of southern Balkans." is quite general and vague while it should be more useful to add some concrete details on the contribution.
A: ABSTRACT HAS BEEN CHANGED

R: Pag. 2, line 1 and 5 = "understanding past climate and its evolution" : this sentence is not linked to the next paragraph where you introduce 'vegetation dynamics'; I suggest to change the first sentence as "understanding past climate changes and vegetation dynamics".
A: DONE

R: Pag. 3, line 7 = smallest Line 27-31= put latin names in italics; line 34 = 'sylvestris'
Pag- 4, line 20 = delete 'only'; line 29 = please, specify the main pollen sum for percentage calculations (all pollen? total land pollen?) Pag 5, lines 15-25 = check, some parts seem methods rather than results
A: DONE

R: D1 = gradual decrease of Artemisia and increase of Quercus = LateGlacial phases; fragmented Pinus = alluvial deposits?
A: AUTHORS AGREE WITH REVIEWER'S INTERPRETATION THAT WILL REPORTED IN THE DISCUSSION SECTION. AT THIS STAGE OF THE MANUSCRIPT WE LIMITED AS POSSIBLE THE INTERPRETATION AND PRESENTED ONLY THE RESULTS

R: Pag. 6, line 32 = Pinus 15% does not indicate a local grow of pine trees
A: THE SENTENCE HAS BEEN CORRECTED

R: Pag. 7, zone D4 = there is the disappearance of Ephedra, decrease of Artemisia, end of Centaurea, presence of Alisma (and sensible decrease of Betula); Galium and Filipendula has the last high values and then will decrease in the next zone = most pollen evidence points to a wet phase, but from your text it seems that this relies on sedimentological data ("According to the sedimentological data, this time period was characterized by increasing humidity in particular during summer.")
A: THE SENTENCE HAS BEEN CHANGED

R: Pag. 7, line 28-29 = "The rising AP% in zone D-3 to D-5 is paralleled by a decreasing trend in the average chain length of vascular plant n-alkanes, also indicating in- creasing arboreal vegetation" = this sentence is not useful here or should be reformulate/completed: practically, you write that 'rising AP' indicates 'increasing arboreal vegetation' and refer to the n-alkanes curve without figure citation (and the reference to published paper is some pages before). I think that here (in the result section) you should point to the increase of Quercus, Abies, Pinus which give a great contribution to this gradual increase matching the rise of wet (and cool?) conditions. In general, in the result section, I suggest to point to your palynological data to obtain palaeoclimatic /palaeoecological inferences because data are strong and clear.

A:  THE SENTENCE HAS BEEN MOVED

R:  Pag. 8, line 9 = cereal traces are present even before; you have 'Cerealia type' pollen grains that may belong to wild species with large pollen – you should mention this in the method section, and therefore probably the term 'Cerealia type' may be more appropriate to this case than others
A:  DONE

R:  Pag. 8, line 24 =" Cereals (>0 %)" ?
A:  THE TYPO HAS BEEN CORRECTED

R:  Pag. 9 Vitis was high even in D6 OJC+Vitis? = I do remark that the OJC group (Fig. 2) shows a clear trend in your diagram (if Olea+Juglans+Castanea are summed up, the curve has a clear increase at around 2500 BP, to which they contribute in this order: first Olea, then Juglans then Castanea) while the scattered curve of Vitis has not the same sharp difference between the 'before' and 'after' the 2500 BP. The Vitis curve is not comparable and has not the same significance of OJC: I cannot agree with your sentence "These four taxa show slightly different behaviours and are sporadically present since the early Holocene." because it is not what we see in the diagram
A:  TEXT HAS BEEN CHANGED ACCORDING TO REFEREE'S INDICATION

R:  Pag. 10 line22 = "Fluctuating presence of coniferous and deciduous taxa in glacial periods is recorded in several lakes" : Fluctuating presence of (coniferous and deciduous) taxa is recorded during the Late Glacial oscillations. Don't you have a Bolling/Allerod – Younger Dryas wet-warm/dry-cold oscillation in your zone D1? Also, of interest, is that you have a sharp increase of AP curve (D2) but then (D3) each tree (Quercus, Rosaceae, Ulmus, Juniperus,. . .) has a gradually increasing curve meaning that this is truly a vegetational (rather than floristic) reply. This gradual increase matches the gradual decline of the steppe of Artemisia+Am/Chenopodiaceae showing a fairly conservative-resilient ecosystem that characterises this area and many lakes you cited.
A: WE USE THE REFEREE SUGGESTION TO ENRICH THE DISCUSSION

R:  Pag 12 lines 1-3 = as you mention Bronze Age, and Neolithic, please, put the relevant millennia in brackets to show the chronology of these phases
A:  AUTHORS ADDED THE INFORMATION WHEN AVAILABLE IN THE REFERENCES. UNFORTUNATELY MANY ARCAHEOLOGAL PAPERS USUALLY AVOID TO BE CLEAR IN THE CHRONOLOGICAL FRAME

R:  "The introduction of Juglans in the Balkans is usually dated at ca. 3000 yr BP (Sadori et al., 2013)." = do you mean that the earlier pollen record is found at c. 3000 BP in another lake? Please, explain and then: "In this frame, the early presence of walnut ". . . in your Dorjan record?
A:  TEXT HAS BEEN IMPORVED

R:  Line 14: "In central Italy, a decrease in humidity, detected soon before 4000 yr BP, is found in low-stand lake levels (Giraudi et al., 2011) and in speleothems (Zanchetta et al., 2016)." Possibly this could be more related to the trends of some pollen curve of your record. I noted that your diagram shows a very similar trend to what I find in the Adriatic core RF93-30: The fall of Quercus ilex type occurred in core RF93-30 at around 4130 cal. B.P., and it is contemporary with both the rise of deciduous oaks (your Q.cerris, I mean) and the thinning of

Abies and Juniperus type (this latter is less evident in your record) " (Mercuri et al. 2012, p. 362). The dryness trend in the marine record, however, also caused a gradual decrease of Fagus

A: WE ARE GRATEFUL TO THE REVIEWER FOR THE SUGGESTION THAT CONFIRMS THE INTERPRETATION OF OUR DATA AND WE ADDED THE QUOTATION TO THE ARTICLE

R: Line 26: I suggest to revise the consideration of Vitis as unambiguously included in the 'cultivated taxa' - look at your Vitis curve

A: TEXT HAS BEEN CHANGED

R: Line 27 = "Mercuri et al. (2013) introduced the OJC (Olea, Juglans, Castanea) sum to estimate the rate of human impact on natural ecosystems" – the sum was firstly cal- culated to follow the development of cultural landscape in central Mediterranean (see also Mercuri 2014 Landscape Ecology, and also for the on site/off site record interpre- tation). The human impact was especially investigated by the API sum (Anthropogenic Pollen indicators; Mercuri et al. 2013 Annali di Botanica) which calculate the percent- age sum of the seven pollen typologies found in archaeological site layers: the API sum indicates the flora which is common (ubiquitarian) near the sites where humans lived.

A: TEXT HAS BEEN CHANGED ACCORDING TO REFEREE'S INDICATION

R: Line 19 = are your pollen grains Cichorioideae or Cichorieae ? according to the paper by Florenzano et al 2015, only Cichorieae have fenestrate pollen

A: WE INCREASE THE DETAIL OF OUR POLLEN IDENTIFICATION THANKS TO THE REFEREE'S INDICATION. THE TEXT HAS BEEN CHANGED. HOWEVER WE DECIDED TO KEEP Cichorioideae AS IT FOR A MORE SIMPLE UNDERSTANDING OF NOT SPECIALISTS

R: Line 22 = You mention that "The strong human activities consisting in livestock farming, fire use, cultivation, overlapped the natural changes of LIA. " but the trends of your pollen curves are not all evidences (again, and also) that there was an impressive resilience in vegetation dynamics of this area?

A: TEXT HAS BEEN CHANGED

R: Line 28 = this is the second time you refer to Paspalum and Phragmites: I know that it is possible to identify Phragmites, but is it also reliable to identify Paspalum (put it in the methods)? If you, however, are referring to the modern vegetation you know from vegetation papers, please specify and change a bit the sentence

A: YES, WE REFER TO MODERN VEGETATION. THE SENTENCE HAS BEEN CHANGED

R: Pag 13, Line 6 = "found also in other sites": mention these sites or delete these words, as you have just said the same thing in the previous sentence. In general, you said several times that you compare data but there are sites that are present only in the figure and not mentioned in the text. A general comment referring to the figure 5 – also outlining the remarkable similarities or dissimilarities – should be added as a conclusive consideration of peculiarities and relevance of Dojran. The different elements visible in the diagram should be distinguished in the final sentences, describing the importance of this off-site record. Also, I suggest incorporating in the conclusions a comment on gradual trends, and resilience

A: WE FOLLOW THE REFEREE SUGGESTION IMPROVING THE TEXT

RESPONSE TO REVIEWER 2

This paper is an important addition to the study of the vegetation history and paleoclimatology of the Balkans in the Holocene. It presents new pollen data and compares to other proxies produced by some of the authors and published previously. The authors' main findings are that there was a rise in arboreal vegetation at the beginning of the Holocene but that it takes a couple of millennia to be definitively attested. This supports the findings of many previous studies from the Mediterranean region. The fact most taxa are present at the bottom of the core suggests a glacial refugia in the catchment. They also find very interesting first signs of human presence ←·5ka (cereal pollen) and then various human impacts through the next few millennia. The methodology and interpretation is robust, there are new findings that are of interest to the community, and it is generally well written with good quality figures, so I recommend this for publication if the following points are considered (especially in the discussion).

While I feel the interpretations are sound and the stories interesting, the authors could expand on the work by comparing their results and findings more to other studies and thinking about the wider implications and underlying climate dynamics that may have caused the observed changes. E.g.:

A: WE THANK THE REFEREE #2 FOR USEFUL AND CONSTRUCTIVE COMMENTS. PLEASE FIND OUR REPLIES BELOW.

1. The authors do compare their results to those from further afield e.g. Italy and Greece. However, there are very similar patterns of slow arboreal pollen increase even further afield e.g. in central and eastern Turkey (Van, Eski) too – may be worth discussing this and how all across the Balkans and eastern Mediterranean this slow increase is being picked up, and what this tells us about what was happening with climate/vegetation change.

A: WE INCLUDED THE SUGGESTED RECORDS TO THE DISCUSSION

2. More discussion of why there is no response in the pollen record to the 8.2ka event would be good – you talk about the lack of response in the pollen record, which is really interesting – but it would be nice to hear some thought on why there was no response, when there was in e.g. Tenaghi Philippon's pollen record (Pross et al., 2009).

A: AUTHORS, FOLLOWING REFEREE'S SUGGESTION, HAVE DISCUSSED MORE IN DETAIL THE 8.2 EVENT

3. "Athanasiadis et al. 2000 provided pollen results for littoral cores from Lake Dojran covering the last 5000 years" – could you compare your pollen record to theirs? Are there differences?

A: WE COMPARED MORE IN DETAIL OUR RECORD WITH THE ATHANASIADIS ONES EVEN IF A DETAILDED CHRONOLOGY IS MISSING AND THE COMPARISON CANNO'T BE MADE IN DETAIL. SOME SILIMARITIES/DISSIMILARITIES ARE ANYWAY EVIDENCED

4. In the results section 4.3, you talk about how several different proxies indicate increasing humidity in this pollen zone, but the diatom assemblage data suggest relatively shallow waters – it would be nice to offer some reasons for this discrepancy (even if these are discussed in the Zhang et al. paper already).

A: THE PARAGPAPH HAS BEEN CHANGED

[revised manuscript text omitted]